# A chromosome-level genome assembly of *Cydia pomonella* provides insights into chemical ecology and insecticide resistance

Fanghao Wan et al.[#]

The codling moth *Cydia pomonella*, a major invasive pest of pome fruit, has spread around the globe in the last half century. We generated a chromosome-level scaffold assembly including the Z chromosome and a portion of the W chromosome. This assembly reveals the duplication of an olfactory receptor gene (*OR3*), which we demonstrate enhances the ability of *C. pomonella* to exploit kairomones and pheromones in locating both host plants and mates. Genome-wide association studies contrasting insecticide-resistant and susceptible strains identify hundreds of single nucleotide polymorphisms (SNPs) potentially associated with insecticide resistance, including three SNPs found in the promoter of *CYP6B2*. RNAi knock-down of *CYP6B2* increases *C. pomonella* sensitivity to two insecticides, deltamethrin and azinphos methyl. The high-quality genome assembly of *C. pomonella* informs the genetic basis of its invasiveness, suggesting the codling moth has distinctive capabilities and adaptive potential that may explain its worldwide expansion.

Correspondence and requests for materials should be addressed to F.W. (email: wanfanghao@caas.cn) or to N.Y. (email: yangnianwan@caas.cn) or to J.R.W. (email: jrwalters@ku.edu) or to F.L. (email: lifei18@zju.edu.cn). [#]A full list of authors and their affiliations appears at the end of the paper.

The codling moth, *Cydia pomonella* (Lepidoptera, Tortricidae), is a wide-spread and highly impactful pest of pome fruit (apples and pears) and walnuts[1]. The larvae of this notorious pest bore into the fruit, causing damage making it unmarketable (Supplementary Fig. 1a). Rates of infestation by *C. pomonella* can reach 80% for apples and 60% for pears in orchards without pest control treatment (Supplementary Fig. 1b)[2]. Records of this species in Greece and Italy from over 2000 years ago suggest it has an origin in Mediterranean Europe[3]. However, its true geographical origin remains unclear and, if associated with the ancient distribution of apples, could be somewhere in the region ranging from Southeast Europe to Asia Minor and across the Caucasus to Central Asia[1]. By 1900, localized populations were documented in Northern Europe, North America, South Africa, South America, and Australasia. In the 21th century, *C. pomonella* increasingly widened its distribution in Europe and North America while also spreading to Africa and Western Asia[4]. Currently, it can be found on six continents (Supplementary Table 1) and imposes severe damages on pome fruit production globally (Supplementary Fig. 1c). Due to its very strong impact on apple crops around the world, more than 20 countries free of this pest maintain a quarantine on this species[5].

The global spread of *C. pomonella* raises key questions concerning which attributes contribute to its success in colonization. How does *C. pomonella* efficiently find food and mates when introduced into a new region? In addition, efforts to control codling moth in recent decades have mainly relied on insecticides, which unfortunately has resulted in high levels of resistance[6,7] and demonstrated this species' striking ability to adapt to acute abiotic stress. What genomic features underlie such rapid adaptation, and might have contribute to the spread of *C. pomonella*?

To catalyze the use of genomics in addressing these questions, we have generated a chromosome-level genome assembly of *C. pomonella* through the combined application of Illumina and Pacific Biosciences (PacBio) sequencing, paired with scaffolding via BioNano and Hi–C. Using this resource, we illuminate the genetic basis of mate and host detection as well as stress resistance. Understanding these biological processes is important to prevent further range expansions and to develop environmentally friendly pest control methods such as mating disruption and sterile insect technique. Meeting such goals would not only tremendously benefit global pome fruit production but would further elucidate causes of the worldwide distribution of many insects.

## Results

**Chromosome-length scaffold assembly of the codling moth.** Our genome assembly strategy employed both Illumina short-read and PacBio long-read sequencing data, with scaffolding informed by both BioNano optical mapping and Hi–C chromosomal contact information. DNA for sequencing was purified from 42 adult females of the Jiuquan strain of *C. pomonella*. This strain was established from specimens collected in Jiuquan city of Gansu province in December 2013 and subsequently maintained on artificial diet in the laboratory. We constructed four paired-end libraries (180, 300, 500, and 800 bp) and three mate-paired libraries (3, 8, and 10 Kb), which were sequenced on the Illumina HiSeq 2000 platform (Supplementary Table 2). This yielded 245.5 Gb of clean data after removing the low-quality reads, corresponding to ~390-fold coverage of the genome, which has a haploid size estimated at ~630 Mb by k-mer analysis and flow cytometry (Supplementary Table 3; Supplementary Figs. 2 and 3). Next, we constructed eight PacBio libraries, which were sequenced on 38 cells using the PacBio Sequel platform, yielding 54.6 Gb of clean data, corresponding to ~86-fold coverage of the genome (Supplementary Table 4).

We obtained 2221 contigs spanning 682.49 Mb with a contig N50 of 862 kb. The assembly was then significantly improved using BioNano optical mapping, yielding 1717 scaffolds spanning 772.89 Mb with a scaffold N50 of 8.9 Mb (Supplementary Table 5). Finally, Hi–C linking information further supported 1108 scaffolds being anchored, ordered, and oriented to give 29 chromosomes (27 autosomes, with Z and W sex chromosomes, Supplementary Table 6), with more than 97% of assembled bases located on the chromosomes (Table 1; Fig. 1a; Supplementary Table 7).

We noticed a large number of gaps were introduced into the chromosomal super-scaffolds after Hi–C scaffolding, which might be caused by high heterozygosity of *C. pomonella*; this situation was also observed in the *Rosa chinensis* genome assembly when assembled with Hi–C scaffolding[8]. To estimate the reliability of genome assembly, we obtained two versions of scaffolds: (1) PacBio scaffolds followed by Hi–C scaffolding and (2) super-scaffolds combining PacBio assembly, BioNano improvement, and Hi–C scaffolding. Aligning these two versions of genome assembly revealed that the two versions showed extremely high collinearity (Supplementary Fig. 4). We used the super-scaffolds for further analysis and evaluated the completeness of the *C. pomonella*

---

**Table 1 Chromosome-level assembled Lepidoptera genomes**

| Features | Cydia pomonella | Trichoplusia ni | Bombyx mori | Spodoptera litura | Melitaea cinxia | Heliconius melpomene |
|---|---|---|---|---|---|---|
| Genome size (Mb) | 772.89 | 368.2 | 431.7 | 438.32 | 393 | 269 |
| Karyotype | 2n = 56 | 2n = 54 | 2n = 56 | 2n = 62 | 2n = 62 | 2n = 42 |
| Number of contigs | 2221 | 26,605 | 15,018 | 13,636 | 49,851 | – |
| Number of scaffolds | 1717 | 6181 | 7397 | 3597 | 8262 | 3807 |
| Number of assembled chromosomes | 27 A + Z + W | 26 A + Z + W | 27 A + Z | 30 A + Z | 30 A + Z | 20 A + Z |
| *Genome assembly quality* | | | | | | |
| Contig N50 (kb) | 862.49 | 621.9 | 15.5 | 68.35 | 13 | 51 |
| Scaffold N50 (Mb) | 8.92 | 14.2 | 3.7 | 0.915 | 0.119 | 0.277 |
| Percentage of scaffolds in chromosomes (%) | 97.48 | 90.62 | 87.30 | 91.08 | 72.45 | 82.68 |
| BUSCO genes (%) | 98.5 | 97.8 | 97.7 | 98.3 | 91.5 | 97.4 |
| *Genomic features* | | | | | | |
| Repeat (%) | 42.87 | 20.5 | 43.6 | 31.83 | 28 | 24.94 |
| G + C (%) | 37.43 | 35.6 | 37.3 | 36.5 | 33 | – |
| *Gene annotation* | | | | | | |
| Number of genes | 17,184 | 14,043 | 14,623 | 15,317 | 16,667 | 12,669 |

*A autosome, Z Z chromosome, W W chromosome*

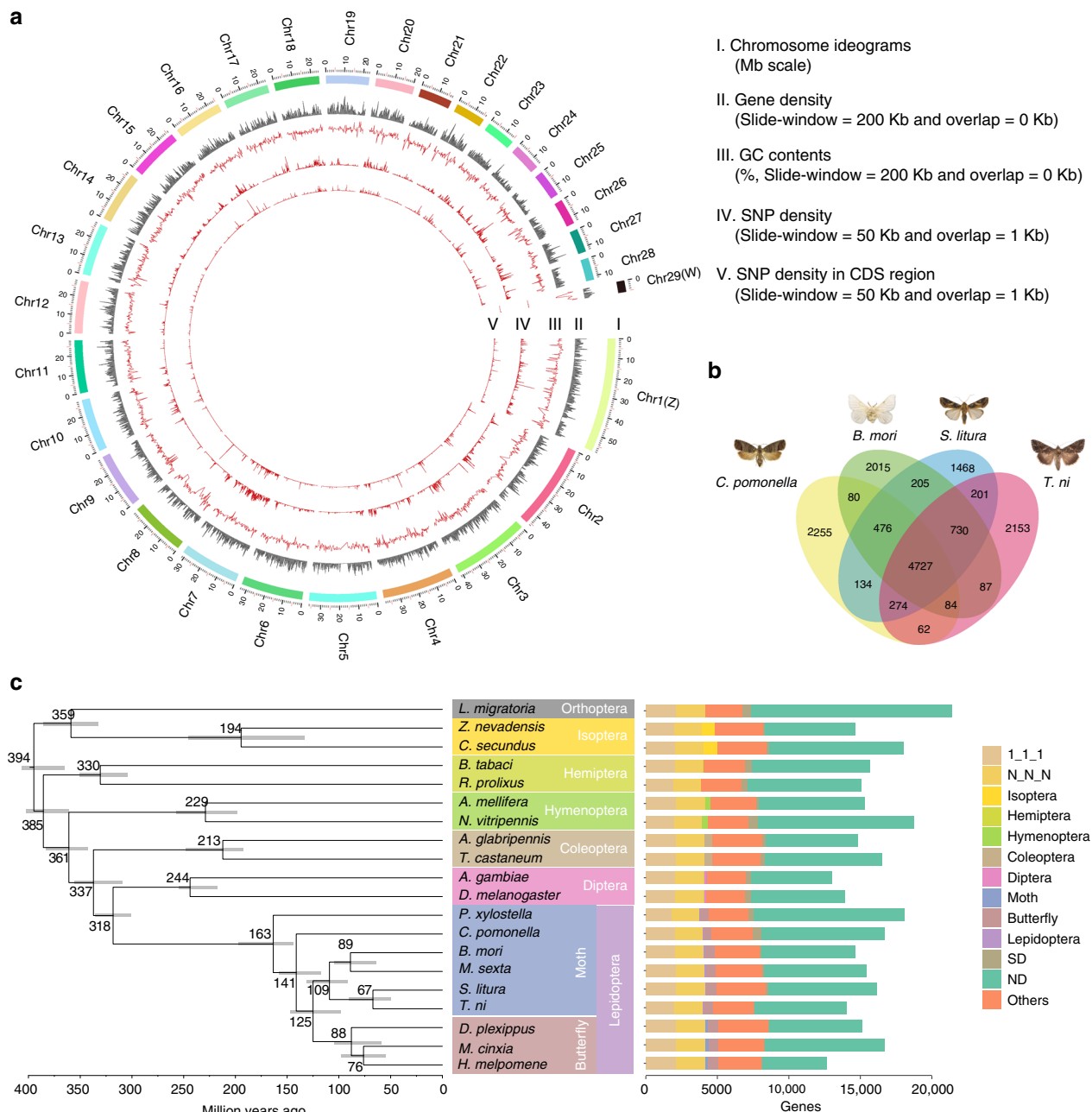

**Fig. 1** Genomic characterization and comparative genomics of the codling moth, *Cydia pomonella*. **a** Circular diagram depicting the genomic landscape of the 29 codling moth chromosomes (Chr1–Chr29 on a Mb scale). The denotation of each track is listed in the center of the circle. **b** The Venn diagram indicates the numbers of treefam annotation families shared among the four Lepidoptera, *C. pomonella*, *Bombyx mori*, *Spodoptera litura*, and *Trichoplusia ni*. Each gene was blasted with the treefam database with the *E*-value 1e−5 and the best hit treefam annotation was selected. **c** Phylogenetic tree and gene orthology of *C. pomonella* with 19 insect genomes. The phylogeny was inferred from 500 strict single-copy genes with 59,621 reliable sites by RAxML maximum likelihood methods employing LG + G model and 100 bootstrap replicates. All nodes received bootstrap support = 100. Divergences were estimated by the PhyloBayes Bayesian method using a relaxed clock with nodes calibration (Supplementary Materials): mean age is given for each note with gray bars indicating 95% posterior densities. Bars giving gene counts are subdivided to represent classes of orthology. 1:1:1 indicates universal single-copy genes, duplication in a single genome and absence less than two moth species. N:N:N indicates other universal genes. SD species-specific duplicated genes, ND species-specific genes. Others indicates all other orthologous groups

genome assembly with the Arthropoda data set of the Benchmark of Universal Single-Copy Orthologs (BUSCO v3)[9], indicating that 98.5% of the gene orthologs were captured (Table 1; Supplementary Table 8). To further validate the genome assembly quality, we sequenced genomic DNA using the Oxford Nanopore platform, yielding ~71 Gb data. More than 99% of these reads mapped to assembly scaffolds, including over 6000 reads longer than 100 Kb

that aligned uniquely and consistently (Supplementary Table 9). In addition, we performed PacBio RNA sequencing and obtained >15,000 consensus transcripts with complete ORFs, of which >93% were mapped to the assembly (Supplementary Table 10). Furthermore, lepidopteran genomes typically exhibit very high levels of synteny[10,11]. Whole-genome alignment of the *C. pomonella* assembly to the chromosomes of the noctuid moth

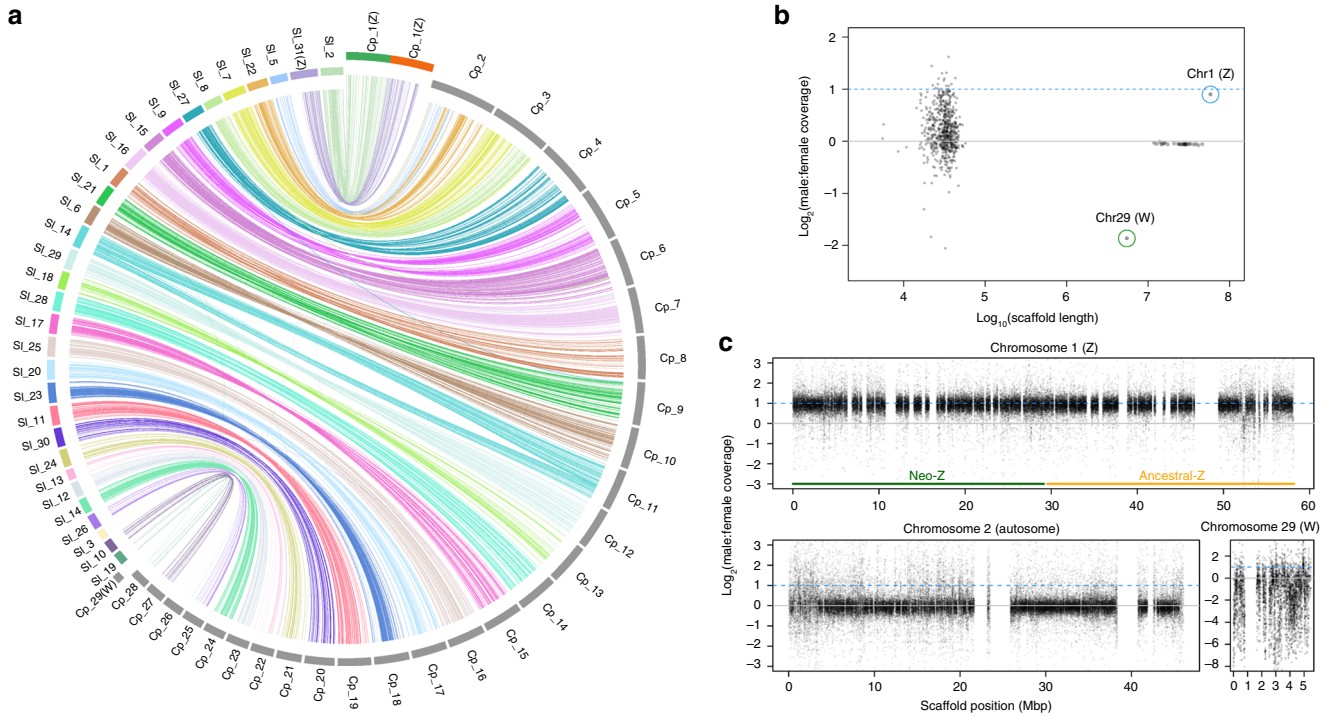

**Fig. 2** Chromosomal identification and evolution of the codling moth, *Cydia pomonella*. **a** Comparative analysis of synteny between *C. pomonella* (Cpom) and *Spodoptera litura* (Slit) chromosomes. **b** Male:female coverage ratios for each scaffold, plotted by scaffold length. Each point represents a single scaffold. The solid gray line is the theoretical expectation for autosomes and the dotted blue line shows the expectation for the Z chromosome. **c** Male:female coverage ratios plotted in 500 bp windows across scaffolds for Z, W, and a representative autosome (Chr 2). Line colors as described in (**b**)

*Spodoptera litura* revealed that chromosomal linkage and ordering of genes are highly conserved[12] (Fig. 2a). All these analyses proved the reliability and completeness of the genome assembly.

**Genome annotation**. In total, 1,692,215 repeat sequences spanning ~341.5 Mb were identified, constituting 42.87% of the codling moth genome (Supplementary Table 11). We used the Optimized Maker-based Insect Genome Annotation (OMIGA)[13] to annotate protein-coding genes, producing 16,997 protein-coding genes in the codling moth genome by integrating the expression evidence from 28 relevant RNA-Seq samples (Supplementary Table 12). We further manually annotated several well-studied gene families proposed to be important in insect adaptation[12], including 85 olfactory receptors (ORs), 65 gustatory receptors, 39 ionotropic receptors, 50 odorant-binding proteins, 28 chemosensory proteins, 136 cytochrome P450s, 47 ATP-binding cassette transporters (ABC transporters), 73 carboxylesterase, 30 glutathione S-transferase, nine nicotinic acetylcholine receptor, two acetylcholinesterase, and one voltage gated sodium channel (Supplementary Tables 13 and 14; Supplementary Figs. 5–7). After removing redundancy, we obtained 17,184 protein-coding genes in *C. pomonella*, which have similar gene features with other lepidopteran genes (Table 1; Supplementary Table 15). Among which, 4727 *C. pomonella* genes have homologous in *Bombyx mori*, *S. litura*, and *Trichoplusia ni* (Fig. 1b). Next, we identified different types of noncoding RNAs (ncRNAs), including 82 small nucleolar RNAs by Infernal and 137,752 Piwi-interacting RNA by Piano[14], 2435 transfer RNAs (tRNA) using tRNAscan-SE[15], 334 ribosomal RNAs (rRNA) using RNAmmer[16], and 217 microRNAs (miRNA) using miRDeep2[17] with

sequencing data from a small RNA library (Supplementary Tables 16 and 17).

We used the OrthoMCL[18] to identify orthologous genes among *C. pomonella* and 19 other insect species covering seven insect orders (Lepidoptera, Diptera, Coleoptera, Hymenoptera, Hemiptera, Isoptera, and Orthoptera). A total of 2124 1:1:1 single-copy orthologous genes and 2051 N:N:N genes were identified (Supplementary Table 18). We inferred a phylogeny and divergence estimate using 500 orthologs (including 452,467 amino acids) concatenated using Gblocks[19] with default parameters (59,621 final amino acids dataset). The lineage to which *C. pomonella* belongs was estimated to diverge from obtectomeran Lepidoptera approximately 141 Mya ago (Fig. 1c).

**Synteny, karyotype evolution, and sex chromosomes**. *C. pomonella* showed a high level of synteny with other lepidopteran genomes (Fig. 2a). Because *S. litura* exhibits the ancestral karyotype of 31 chromosomes[12], this comparison also provides information on the karyotype changes that occurred in the lineage leading to *C. pomonella*. Previous cytogenetic analysis established that *C. pomonella* has 27 autosomes, a female specific W chromosome, and a neo-Z chromosome arising from a Z-autosomal fusion involving an autosome homologous to chromosome 15 in *Bombyx mori*[20,21]. Comparison to *S. litura* confirms this fusion event, which gave rise to the largest *C. pomonella* chromosome. It also reveals two additional fusion events involving chr2 (fusion of Chr5 and Chr22 of *S. litura*) and chr3 (fusion of Chr7 and Chr8 of *S. litura*) (Fig. 2a), thus resolving each of the three fusion events that occurred to produce the n = 28 karyotype observed in *C. pomonella* and related olethreutin moths[22].

Using resequencing data from three males and three females, we assessed sex-specific patterns of sequencing coverage across

scaffolds and confirmed the presence of the Z chromosome and a portion of the W in our assembly (Fig. 2b). All but two chromosomal-length scaffolds showed equal coverage between sexes, as expected for autosomes. The largest scaffold (chr1) yielded twofold greater male coverage, as expected for the Z chromosome. This twofold difference is consistent across both the ancestral and neo-parts of the Z (Fig. 2c), indicating very little remaining sequence homology, if any, between the neo-Z segment and the current W sequence, as suggested by previous cytogenetic work[20]. In contrast, the chr29 scaffold showed a strongly female-biased coverage ratio, indicating it represents W-linked sequences (Fig. 2c). The pattern of male:female coverage is much more variable across the chr29 scaffold than for other chromosomes (Fig. 2c). This likely reflects the abundance of transposable elements (TEs) on the W which are likely to collect read mappings from homologous TEs in other regions of the genome.

Cytogenetic analysis revealed little evidence of shared sequences between the Z and W, suggesting loss or nearly complete degeneration of the neo-W chromosome segment[20,22]. Similarly, our efforts to detect patterns of collinearity or gametologs between the assembled W segment and the Z chromosome did not yield positive results (Supplementary Materials Section 4.4; Supplementary Fig. 8). Thus, primarily through the absence of any strong detectable homology between the Z and W sequences in the *C. pomonella* assembly, we confirm the substantial degradation or loss of the female-limited homolog of the neo-Z in the *C. pomonella* lineage. However, cytogenetic data indicate that the W chromosome is approximately the same size as the Z[22], while the chr29 scaffold is only about 1/10 the size the chr1 scaffold, suggesting that the chr29 scaffold represents only a fraction of the entire W chromosome. Accordingly, a more comprehensive assembly of the W chromosome is needed to robustly address the fate of the neo-W because there may be as yet unassembled portions of the W chromosome that could show homology to the neo-Z.

We further explored various sequence characteristics of chr29 relative to the rest of the genome. The proportion of GC bases is slightly elevated compared to other chromosomes (Supplementary Fig. 9). As is typical of non-recombining hemizygous chromosomes[23], lepidopteran W chromosomes are highly degenerate, being gene-poor and repeat-rich. Chr29 does indeed appear to be gene-poor: we detected no chr29 protein-coding genes that appear to be anything other than TEs. However, results from repeat masking do not indicate notably greater repeat content than other chromosomes, though the structure and composition of W-linked repeats do appear distinct (Supplementary Fig. 10). They are considerably fewer in total number but have greater average length compared to the other chromosomes. Also, the W hosts a notably larger proportion of long terminal repeat and DNA transposons compared to the other chromosomes (Supplementary Fig. 11). A de novo clustering analysis of sex-specific sequencing data identified at least two repeats that were significantly enriched in females, and presumably W-linked (Supplementary Fig. 12; Supplementary Table 19).

**OR3 duplication enhances the ability to locate food and mates**. In insects, the chemosensory system mediates many behaviors such as locating food, shelter, mates, and oviposition sites[24–29]. It thus plays an important role in determining the invasiveness of insects[30], particularly for oligophagous species like *C. pomonella*. Studies of chemosensation in codling moth have established that both sexes are strongly attracted to the plant volatile pear ester, which can also substantially enhance the male-specific response to codlemone, the major female-produced sex pheromone[31].

However, the mechanism of synergy between pear ester and codlemone in male response remains elusive, motivating our efforts to further characterize the repertoire of chemosensory genes in *C. pomonella*.

Our high-quality draft genome provided the novel opportunity to comprehensively annotate and analyze relevant genes. We identified a total of 85 *OR* genes in the *C. pomonella* genome and performed a phylogenetic analysis, finding an expansion of the pheromone receptor cluster in *C. pomonella* (Supplementary Fig. 6). Furthermore, by examining the chromosomal locations of all *OR* genes (Supplementary Fig. 5), we found two copies of *OR3*, namely *CpomOR3a* and *CpomOR3b* (Supplementary Table 13), both with the same gene length and the same exon–intron structure, forming a tandem repeat on chromosome 17 with an intergenic interval of 9812 nt (Fig. 3a; Supplementary Fig. 13). In a previous study based on the analysis of an antenna transcriptome, only one copy of *CpomOR3* was identified, whose protein product detects pear ester[32]. To make sure the *OR3* duplication is fixed and not segregating (with some moths having one copy and some two), we confirmed the presence of the duplication in all resequenced moths.

Having identified this duplication, we subsequently addressed whether it contributes to the enhanced ability of the codling moth to detect pear ester. First, we confirmed the expression of both *CpomOR3a* and *CpomOR3b* by gene specific-reverse transcription-polymerase chain reaction (PCR). Then, we calculated fragments per kilobase million (FPKM) values from 24 RNA-Seq transcriptome datasets from various tissues and showed that *CpomOR3b* was expressed only in adult antennae (male FPKM = 1050.17, female FPKM = 4014.13), while *CpomOR3a* was expressed in adult antennae of both male (FPKM = 41170.2) and female (FPKM = 88916.8), as well as in adult heads (FPKM = 14771.4–68715.2) and larval heads (FPKM = 7627.82) (Fig. 3b). Compared to the other *CpomORs*, the OR3 duplicates were among the most highly expressed in adult female antennae. Fluorescence in situ hybridization on *C. pomonella* adult antennae showed that *CpomOR3a* and *CpomOR3b* were expressed mainly in different but adjacent neuronal cells within the same sensillum, although some exclusive expression of either *CpomOR3a* or *CpomOR3b* could also be detected in other non-colocalized neurons (Fig. 3c; Supplementary Table 20; Supplementary Fig. 14). Moreover, *CpomOR3a* and *CpomOR3b* were consistently co-expressed with *CpomORco*, the obligatory co-receptor of ORs[33], in *C. pomonella* adult antennae (Fig. 3c). These results suggested that these two copies have distinct expression patterns, inferring that they might have underwent neofunctionalisation and acquired divergent functions.

Because CpomOR3a has been reported as the putative receptor of pear ester, we wondered if CpomOR3b, which presents 89% sequence identity with CpomOR3a, would also detect this compound and contribute to the high sensitivity of the codling moth to this chemical cue. We thus co-expressed *CpomOR3a* or *CpomOR3b* together with *CpomORco* in *Xenopus* oocytes and used a two-electrode voltage clamp to record each protein's response to pear ester and other chemicals. We found that both copies were functional with similar response spectra. CpomOR3a and CpomOR3b were strongly tuned to pear ester, but both also responded to the sex pheromone codlemone (Fig. 4a). Next, we knocked down *CpomOR3a* and *CpomOR3b*, either separately or simultaneously, by injecting siRNAs in late pupae (Supplementary Table 21). Quantitative PCR (qPCR) showed that each paralog (*CpomOR3a* or *CpomOR3b)* was successfully and specifically knocked down without influencing the alternative paralog's expression (Fig. 4b; Supplementary Table 22). After emergence at 72 h post injection, we assayed the electrical activity of the whole antennae using electroantennography (EAG). EAG

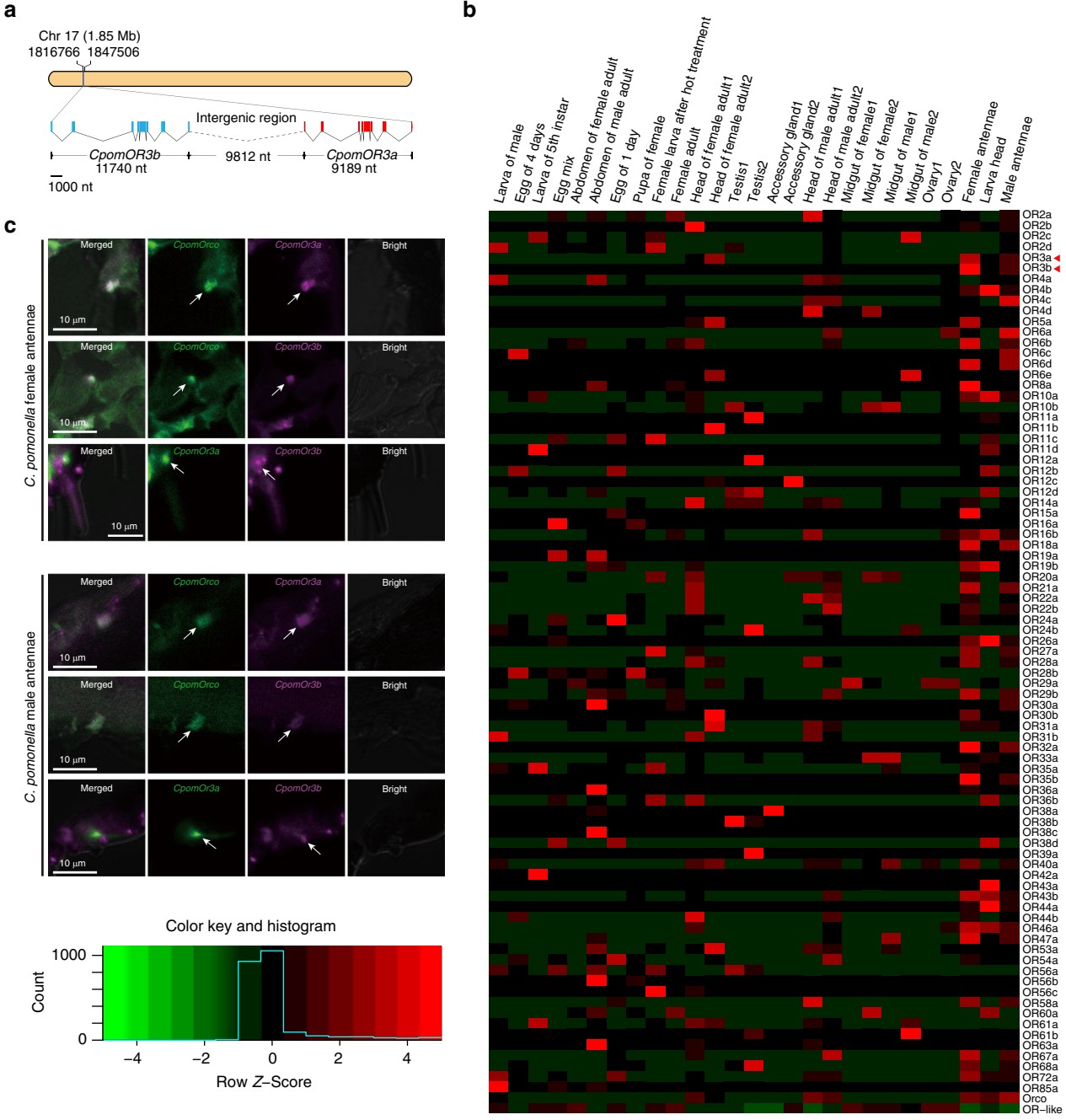

**Fig. 3** Structure and expression profiling of *CpomOR3a* and *CpomOR3b* in the codling moth, *Cydia pomonella*. **a** Exon–intron organization and chromosome location of *CpomOR3a* and *CpomOR3b* genes. The exon–intron organization of each gene was determined by sequence comparison between genomic sequences and putative cDNA sequences with BLASTN. The exons are shown as boxed regions (*CpomOR3b*: blue; *CpomOR3a*: red). The angled solid lines between boxes indicate the introns, while the angled dotted lines indicate the intergenic region. **b** Expression profiling of *CpomOR3a* and *CpomOR3b* in different tissues of *C. pomonella*. Estimation of abundance values determined by read mapping. Green indicates no to low expression, black indicates low to moderate expression, and red indicates moderate to high expression. Each data block shows the scaled *z*-score of FPKM value of the corresponded tissue/ organ. The NCBI SRA accession numbers of all used transcriptomes were given in Supplementary Table 27. **c** Co-expression patterns of *CpomOR3a*, *CpomOR3b*, and *CpomORco* in *C. pomonella* antennae. Two-color FISH was used to label each pair of genes by probes with either digoxigenin (green) or biotin (purple). Upper panels: Female antennae expression. Lower panels: Male antennae expression. For both sexes, row 1: co-expression of *CpomORco* and *CpomOR3a*; row 2: co-expression of *CpomORco* and *CpomOR3b*; row 3: separated expression of *CpomOR3a* and *CpomOR3b*. Source data are provided as a Source Data file

showed that the male responses to pear ester and codlemone were impaired in all siRNA-treated strains when compared with the negative control group treated with siGFP (siRNA designed based on the sequence of green fluorescent protein). In contrast the female's responses to pear ester were impaired only when both *CpomOR3a* and *CpomOR3b* are knocked down (Fig. 4c). Further, we analyzed the behavioral responses of siRNA-treated adults to pear ester or codlemone. The Y-tube assays indicated that simultaneously silencing *CpomOR3a* and *CpomOR3b* significantly impaired the ability of adult moths to trace pear ester in both

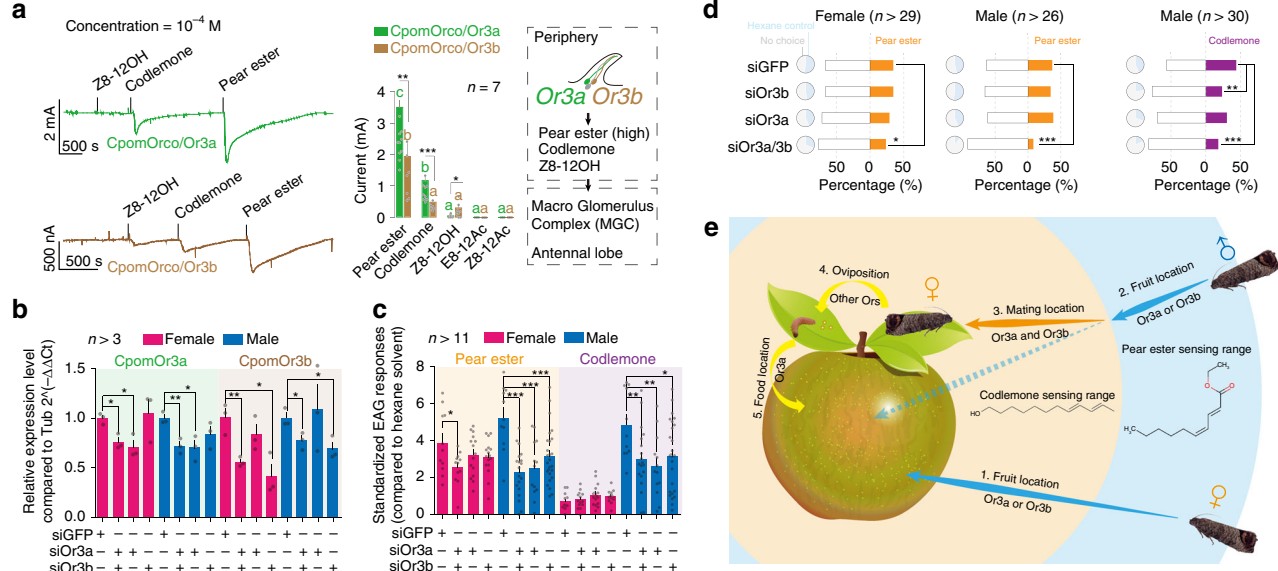

**Fig. 4** Functional demonstration of *CpomOR3a/CpomOR3b* in pear ester and codlemone olfactory reception of the codling moth, *Cydia pomonella*. **a** Comparison of *CpomOR3a* and *CpomOR3b* responses to different chemical components with *Xenopus* expression system. Example traces of *CpomOR3a/CpomORco* and *CpomOR3b/CpomORco* injected *Xenopus* oocytes are shown respectively, with current elicited by Z8-12OH, codlemone, and pear ester solutions at $10^{-4}$ M concentration. Lower case letters indicate significant differences (ANOVA and Tukey HSD, $F_{4, 45} = 161.8$, $P < 0.0001$), or *CpomOR3b/CpomORco* injected oocytes (ANOVA and Tukey HSD, $F_{4, 45} = 22.3$, $P < 0.0001$). * indicates significant differences (Student's t test, pear ester: $t_{15} = 3.6$, $P = 0.0027$; codlemone: $t_{15} = 4.2$, $P = 0.0008$; Z8-12OH: $t_{17} = 2.5$, $P = 0.0235$). Error bars indicate + SEM. (**b**) Quantitative PCR tests for RNAi strains. Data were calculated based on the $2^{-\Delta\Delta Ct}$ method with normalization to *Cpomβ-tubulin*. * indicates significant influences (Student's t test, *CpomOR3a*, female, siOR3a/b: $P = 0.0118$, siOR3a: $P = 0.0181$; male, siOR3a/b: $P = 0.0019$, siOR3a: $P = 0.0273$ (one-tail); *CpomOR3b*, female, siOR3a/b: $P = 0.0090$, siOR3b: $P = 0.0171$; male, siOR3a/b: $P = 0.0284$, siOR3b: $P = 0.0035$). Error bars indicate + SEM. **c** Electroantennogram (EAG) results with RNAi strains to pear ester and codlemone. * indicates significant decrease (Student's t test, pear ester, female, siOR3a/siOR3b: $t_{21} = 2.4$, $P = 0.028$; pear ester, male, siOR3a/siOR3b: $t_{28} = 5.3$, $P < 0.0001$, siOR3a: $t_{22} = 4.1$, $P = 0.0004$, siOR3b: $t_{35} = 3.7$, $P = 0.0007$; codlemone, male, siOR3a/siOR3b: $t_{28} = 2.8$, $P = 0.009$, siOR3a: $t_{22} = 3.1$, $P = 0.005$, siOR3b: $t_{35} = 2.5$, $P = 0.018$). Error bars indicate + SEM. **d** Y-tube assays of RNAi strains. Distributions of counts among choosing of tested chemicals, hexane, and no choice were compared between each injected strain with siGFP strain by chi-square test. Bar charts indicate proportions of counts between positive choice (either codlemone or pear ester) and negative (no choice and hexane control). * indicates significant differences between current treatment with siGFP strain (pear ester, female, siOR3a/siOR3b: $\chi = 9.5$, $P = 0.0084$; pear ester, male, siOR3a/siOR3b: chi $= 24.4$, $P < 0.0001$; codlemone, male, siOR3a/siOR3b: $\chi = 47.4$, $P < 0.0001$, siOR3b: $\chi = 10.5$, $P = 0.0053$). **e** Schematic illustration showing predicted *OR3a* and *OR3b* mediated behaviors in *C. pomonella*. Step 1: female adults locate fruits and find mating sites on leaves. Step 2: male adults locate fruits as potential mating sites from a long distance. Step 3: male adults locate females in a close distance via codlemone sensing process. Steps 4 and 5: oviposition of females and food location of newly hatched larvae via *OR3a* detection of pear ester. Source data are provided as a Source Data file

sexes and codlemone in males. In addition, silencing *CpomOR3b* alone significantly decreased tracing ability of male *C. pomonella* toward codlemone (Fig. 4d).

Previous electrophysiological studies suggest that olfactory sensory neurons (OSNs) detecting codlemone are housed in sensilla trichodea[34]. Since we showed that *CpomOR3a* and *CpomOR3b* responded to codlemone, one might expect them to be located in trichodeal OSNs. However, their expression pattern in adjacent neurons in adult antennae and their response profile are more consistent with OSNs previously described in sensilla auricillica[35,36], suggesting they are more likely expressed in auricillic OSNs. Our results thus suggest the occurrence of a pheromone-specific pathway via ORs (still unknown) expressed in sensilla trichodea and a pheromone/pear ester pathway via *OR3a* and *OR3b* expressed in sensilla auricillica. With these two pathways, *C. pomonella* would have evolved a specially enhanced chemosensory system to efficiently locate food and mates (Fig. 4e).

**GWAS identifies SNPs associated with insecticide resistance.** Application of chemical insecticides is the main method used for controlling codling moth. Unfortunately, this species has developed high levels of resistance worldwide to numerous insecticides[37]. Understanding the genetic mechanisms underlying insecticide resistance is important for developing efficient and sustainable pest management methods.

Main insecticide resistance mechanism in *C. pomonella* relies both on an increased activity of detoxification enzymes and on decreased sensitivity of target proteins to insecticides[38,39]. We identified 667 genes in the codling moth previously reported to be potentially involved in insecticide resistance, including 434 detoxification genes, 45 insecticide target genes, 124 cuticle genes, 47 ABC transporters, and 12 aquaporins (Supplementary Data 1). We focused on the analysis of cytochrome P450 monooxygenase genes because previous biochemical studies suggest that greater hydrolytic P450s activity conferred resistance to a large spectrum of chemical insecticides in *C. pomonella*[40]. Figure 5 shows the distribution of 146 P450 genes across the genome. There were 16 gene clusters including three or more P450 genes. The largest cluster (in Chr20) having 11 P450 genes, including three *CYP6AE* genes. P450 gene clusters have been reported to be involved in insecticide resistance in cotton bollworm *Helicoverpa armigera*[41] and rice stem borer *Chilo suppressalis*[42]. The high number of P450 gene clusters observed here indicate that *C. pomonella* may have enhanced abilities to cope with phytochemical or synthetic toxins.

To identify genetic changes conferring insecticide resistance, we resequenced six individuals from each of three strains (S, Raz,

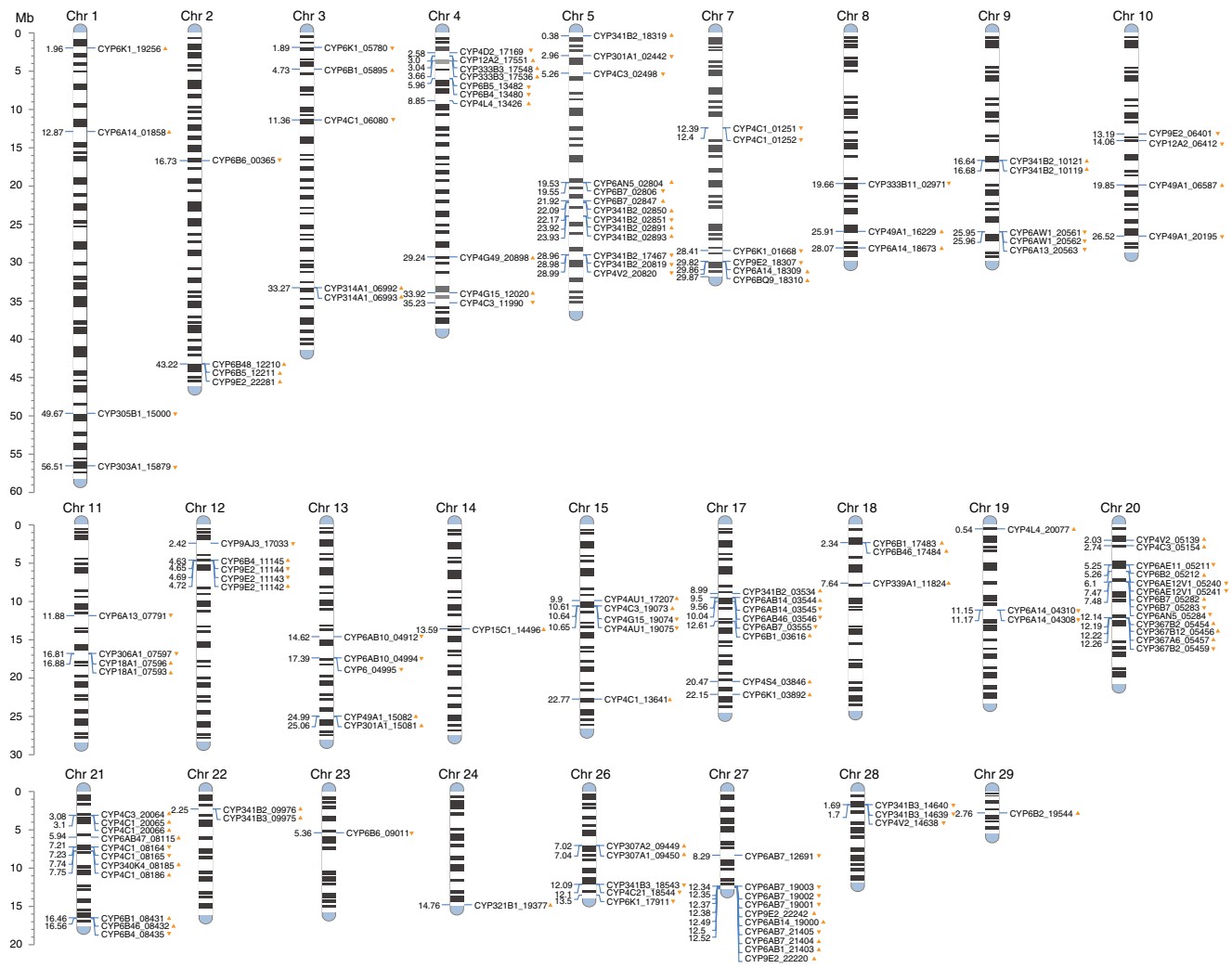

**Fig. 5** The genomic positions of P450 genes in the codling moth, *Cydia pomonella*. The predicted 146 P450 genes were mapped to the genome and the 5′ UTR of 69 P450 were successfully amplified with RACE strategy. The distribution analysis of all 146 P450 genes showed that there are 16 gene clusters which have three or more P450 genes

and Rv) reared on an artificial diet in INRA Avignon, France[43]. The S strain is susceptible to insecticides and has not been exposed to any insecticide since 1995. The Raz strain has been selected for insecticide resistance since 1997 by exposing larvae to azinphos-methyl (375 mg L$^{-1}$); it shows 7-fold resistance to azinphos-methyl and 130-fold resistance to carbaryl in comparison with S[44]. The Rv strain has been selected since 1995 by exposing larvae to deltamethrin (2 mg L$^{-1}$) and shows 140-fold resistance to deltamethrin in comparison with S[45]. The Rv and S strains were selected from the same population collected in an apple orchard at Les Vignères of south-eastern France. The Raz strain comes from a population collected in an apple orchard closed to Lerida, Spain[45].

Six individuals were randomly selected from each strain, and each individual was sequenced at ~40× coverage, yielding a total of 474.6 Gb of data (Supplementary Table 23). We performed a genome-wide association study (GWAS) to identify insecticide-resistance associated single-nucleotide polymorphism (SNPs). When comparing between S and Raz strains, we identified 109 SNPs (nonsynonymous or synonymous) with significantly different allele frequencies located in exonic regions of the abovementioned 667 resistance-associated genes (Fig. 6a; Supplementary Table 24). When comparing S and Rv strains, we identified 242 significantly differentiated SNPs (nonsynonymous

or synonymous) located in exonic regions of resistance-associated genes, of which 18 SNPs were found in both Raz and Rv (Fig. 6b). For 11 of these SNPs, we further assayed tens of individuals from each strain via Sanger sequencing and confirmed seven of them show fixed differences between the S strain and either Raz or Rv strains (Fig. 6c; Supplementary Table 25). Among these confirmed SNPs that have caught our attention are mutations in muscarinic receptors (*mAChR*), octopamine beta receptors and *CYP6B2* P450 genes, which, to best of our knowledge, have never been reported to be involved in insecticide resistance in any Lepidoptera species before (Fig. 6c).

Since P450-based resistance primarily reflects changes in gene expression and presumably results from mutations in regulatory regions, we annotated the 5′UTRs of these genes via rapid amplification of cDNA ends (RACE). Of 136 P450 genes annotated in *C. pomonella*, we obtained the 5′UTR sequences of 69 P450 genes, which we mapped to scaffolds to determine the transcription starting sites (TSS) and promoter regions (−300 bp to +100 bp corresponding to TSS). Among the 136 P450 genes, GWAS analysis identified 128 and 203 SNPs that differentiated the S strain from the Raz or Rv strain, respectively. In the promoter regions of 69 of P450 genes, there were nine and ten SNPs that differentiated the S strain from the Raz or Rv strain, respectively. Notably, we found three SNPs associated with both

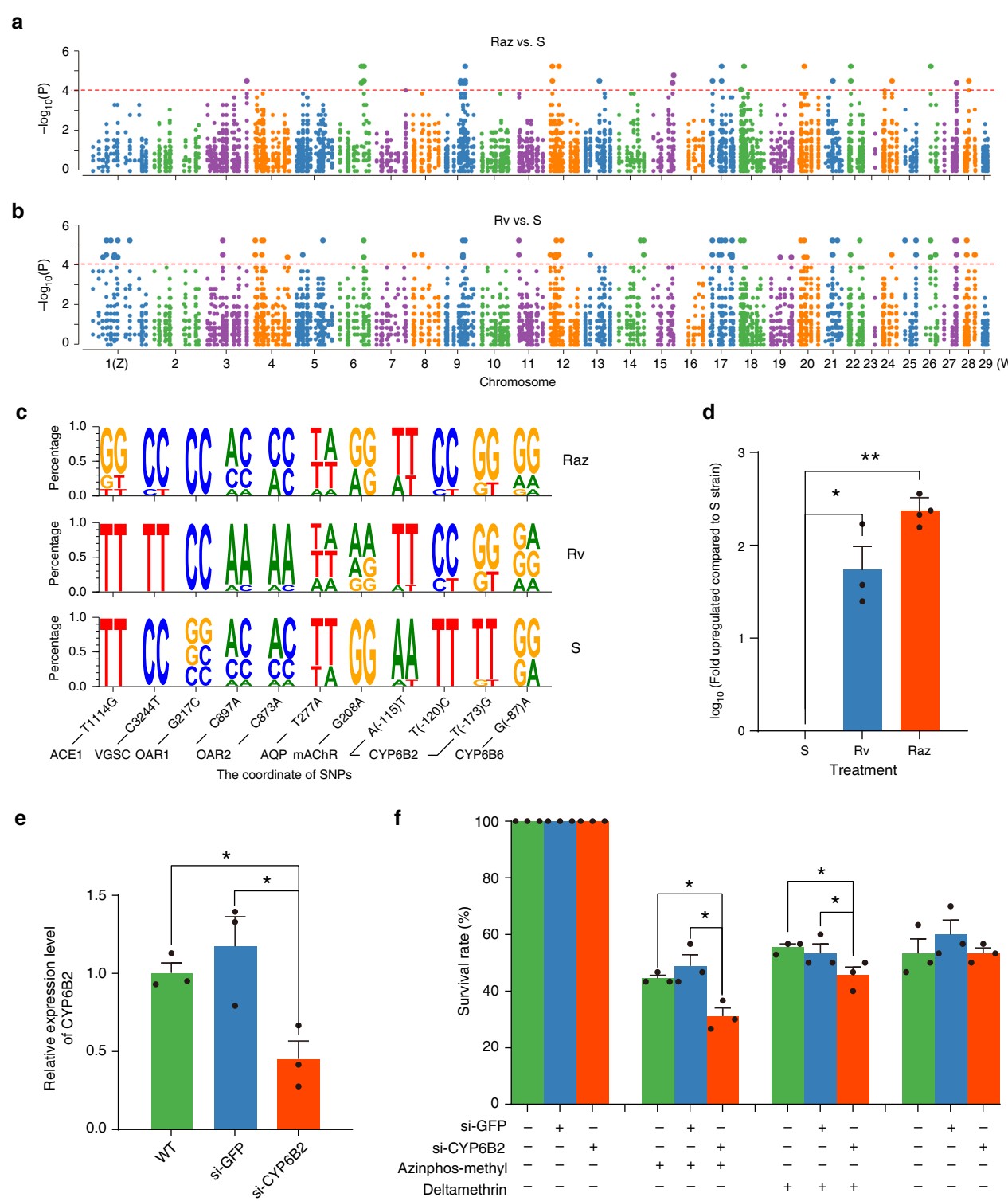

Raz and Rv strains in the promoter of *CYP6B2*: A52T: A (−52)T, T(−57)T, and T(−110)G (gene ID: CPOM05212) (Fig. 6c). Using qPCR, we estimated the expression of *CYP6B2* in the three strains; the results showed that this gene is constitutively overexpressed in the two resistant strains (241.4-fold in Raz and 77.3-fold in Rv) compared with the S strain (Student's *t* test, Rv: *P* = 0.0102, Raz: *P* = 0.0002) (Fig. 6d), suggesting that these SNPs play a role in expression regulation of *CYP6B2*.

Furthermore, to verify that *CYP6B2* expression level is indeed linked to insecticide resistance, we next knocked down *CYP6B2*

by injecting siRNA into the fourth-instar larvae of the Jiuquan strain. The expression level of *CYP6B2* decreased by 55% at 48 h after injection of si*CYP6B2* compared with the negative control (Fig. 6e). We then used LC$_{50}$ concentrations of azinphos methyl (103.50 mg/L), deltamethrin (3.55 mg/L), and imidacloprid (35.35 mg/L) to treat the RNAi individuals. The survival rates of the larvae treated with azinphos methyl and deltamethrin in the si*CYP6B2* group were 31.1% and 45.6%, respectively, which were significantly lower than those of the negative control group treated with siGFP or the group without any treatments (Student

**Fig. 6** Genes involved in insecticide resistance revealed by genome data of the codling moth, *Cydia pomonella*. SNPs in 667 genes involved in the azinphos methyl (**a**) and deltamethrin (**b**) resistance represented by Manhattan plot. The vertical axis shows the *P*-value and the horizontal axis indicates the Z, W, and 27 autosomes. The red horizontal line represents the genome-wide significance threshold ($P = 1 \times 10^{-4}$). **c** The SeqLogo plots show the frequency of diploid genotypes, which were significantly different between resistant strains (Raz, resistance to azinphos methyl; and Rv, resistance to deltamethrin) and the susceptible strain (S); the primer pairs used are given in Supplementary Table 25. Ten individuals of each strain were used for the confirmation. **d** qPCR analysis of the expression level of *CYP6B2* (ID: CPOM05212) in Raz and Rv insecticide-resistant strains in comparison with the insecticide susceptible strain S. The transcript abundance of *CYP6B2* was 241. Fourfold higher in Raz strain and 77.3-fold higher in Rv strain compared with that of the sensitive strain S. * indicates significant influences (Student's *t* test, Rv: $P = 0.0102$, Raz: $P = 0.0002$). This gene was constitutively highly expressed in two resistant strains. Error bars indicate + SEM. **e** The expression levels of the *CYP6B2* were significantly reduced after siRNA injection. * indicates significant influences (Student's *t* test, Azinphos-methyl, WT: $P = 0.0135$, siGFP: $P = 0.0318$). WT indicates individuals without any injection. Error bars indicate + SEM. **f** The survival rate of *C. pomonella* fourth instar larvae treated with LC$_{50}$ concentrations of azinphos methyl (103.50 mg/L), deltamethrin (3.55 mg/L), and imidacloprid (35.35 mg/L) after injection of siRNA. * indicates significant influences (Student's *t* test, Azinphos-methyl, WT: $P = 0.0132$, siGFP: $P = 0.0232$; Deltamethrin, WT: $P = 0.0334$, siGFP: $P = 0.0409$ (one-tail)). Thirty individuals were used for each treatment. Error bars indicate + SEM. Source data are provided as a Source Data file

*t* test, $P < 0.05$) (Fig. 6f), suggesting that knocking down *CYP6B2* significantly increased the sensitivity to deltamethrin or azinphos methyl. The larval sensitivity to imidacloprid, however, was not significantly affected after knocking down *CYP6B2*, suggesting that it is not involved in metabolizing imidacloprid. Taking all evidence together, these results demonstrate that *CYP6B2* plays a critical role in conferring resistance to these two widely used insecticides.

## Discussion

We have generated a high-quality genome assembly of the codling moth *C. pomonella* by combining distinct sequencing strategies, to yield twenty-nine chromosome-level scaffolds, including the Z and W chromosomes. To the best of our knowledge, this is the first assembly yielding chromosome-level scaffolds without linkage mapping for lepidopteran insects sequenced from whole organisms. Another recent example of such a highly contiguous assembly is the 28 chromosomes of the noctuid moth, *Trichoplusia ni*. However, this last project sequenced Hi5 germ cell lines, which were shown to have substantial chromosomal rearrangements and duplications relative to the organismal karyotype[46,47]. Given the known differences in karyotype and heterozygosity that arise in cell lines, the assembly of *C. pomonella* reported here, generated from tissues, reflects an important milestone in genome assembly for lepidopteran insects. The high heterozygosity typical of lepidopteran insects has long impeded efforts to obtain such high-quality genome assemblies[48]. As demonstrated here, by employing long-read sequencing (e.g., PacBio), high throughput physical mapping (e.g., BioNano), and chromatin confirmation assays (e.g., Hi–C), it is feasible to obtain a chromosome-level scaffold assembly, even for notably large, repetitive, and heterozygous genomes such as *C. pomonella*'s. This result heralds a new era of high-quality insect genome assembly without additional data from linkage mapping, which can often be slow, difficult, or impossible for many species.

The genetic basis underlying the global spread of invasive insects remains an outstanding question. Invasiveness may contribute to increasing globalization of a species. Among insects, previous research has suggested that some gene families have been associated with invasiveness of the Mediterranean fruit fly (*Ceratitis capitata*)[49] and the red imported fire ant (*Solenopsis invicta*)[50]. In another example, it has been determined that metabolic plasticity enables the Asian longhorned beetle (*Anoplophora glabripennis*) to employ diverse plant host species and thus contributes to the highly invasive nature of this pest[51]. However, *C. pomonella* is an oligophagous insect and the mechanisms of its successful spread in the past 50 years remains elusive. The genome biology of *C. pomonella* elucidates some key features that may contribute to this. First, our high-quality

assembly revealed without ambiguity that the response of codling moth to pear ester is linked to an OR gene duplication. As both receptors also responded to the sex pheromone codlemone, the duplication event of *OR3* significantly enhances the ability to locate not only food but also mates, which are likely to be crucial traits in the early stage of dispersal and population establishment. Thus, this *OR3* duplication likely contributes substantially to the rapid global spread of *C. pomonella*. It is an unusual finding that a single receptor (and its duplicate) is able to detect compounds involved in mate and oviposition site detection, since most moth pheromone receptors are specifically tuned to pheromone only. Such atypical ORs have been recently described in the oldest lineages of moths (the nonditrysian moths), suggesting that pheromone receptors evolved from receptors tuned to plant volatiles[52]. We report here additional evidence from a ditrysian species, supporting this hypothesis.

Codling moth has substantially adapted to abiotic stresses, as exemplified by the rapid evolution of resistance to various insecticides in natural populations. When comparing our data with previous work on the dengue mosquito *Aedes aegypti*[53], we observed the same proportions of candidate genes in the selected gene families. For instance, cytochrome P450 genes represent 27% and 24% of all candidate genes in *C. pomonella* and *A. aegypti*, respectively. Hundreds of SNPs were identified in candidate genes known to be associated with insecticide resistance. We presented a series of evidence from transcriptomics, gene expression analysis, and RNAi knockdown to suggest that at least three SNPs participate in upregulation of *CYP6B2* expression in resistant strains and thus insecticide resistance. In addition, we also identified thousands of significantly differentiated SNPs in 1778 genes in Raz and 3619 genes in Rv that have not been previously implicated in insecticide resistance (Supplementary Table 26). While it is likely most of these SNPs do not play any role in resistance, this set of genetic differences between strains represents a substantial resource to screen new candidate genes and to discover novel mechanisms involved in insecticide resistance.

In summary, we provide insights into the genetic bases of enhanced chemical sensory sensitivity and potent adaptive ability of codling moth as a worldwide destructive herbivore. The chromosome-level genome assembly will facilitate future genetics studies on the adaptation of codling moth to global agriculture changes and support the development of sustainable strategies for pest control.

## Methods

**Insects.** The *C. pomonella* were collected at Jiuquan city, Gansu province in December 2013 (Jiuquan strain), and then maintained by an artificial diet in the laboratory of the Chinese Academy of Inspection and Quarantine. The insectarium

environment was set at 25 ± 1 °C and 75 ± 5% relative humidity on a photoperiod (Light: Dark = 14:10). The experiments in this work have received ethical approval from the Institute of plant protection, Chinese Academy of Agriculture Science, Beijing, China.

**Genome sequencing**. Genomic DNA was extracted from 42 fifth instar female larvae of an inbred Jiuquan strain which was maintained by sibling mating for six generations. To decrease the risk of nonrandomness, we built different insert sizes libraries. All libraries were sequenced by using Illumina HiSeq 2000 101PE platform. In total, we 245.5 Gb clean data were maintained for genome assembly (Supplementary Table 2). We also generated 54.57 Gb data sequenced for 38 cells by the PacBio RS II sequencing platform at the Annoroad Gene Technology Co. Ltd. (Supplementary Table 4).

**Genome assembly**. The draft genome was assembled using the raw reads of the PacBio and Illumina sequencing platform. We used different methods in combining PacBio and Illumina data to assemble the draft genome and compared the results of different methods, and finally chose the method using PacBio to assemble the frame of the draft genome scaffolds and then polish and improve the scaffolds with Illumina clean reads. To assemble the draft genome scaffolds from the PacBio reads, we used the Falcon v0.3.0 software[54]. Then, we used the Redundans[55] software to remove redundant scaffolds from the assembly and generate a non-redundant assembled genome. Finally, the illumina data were generated to correct the genome assembly by the Pilon software[56].

**BioNano**. To obtain a high-quality genome assembly, the BioNano next-generation mapping system was used. Scaffolding of the contigs/scaffolds with optical mapping was performed using the Irys optical mapping technology (BioNano Genomics) at the Annoroad Gene Technology Co. Ltd. The IrysView (BioNano Genomics) software package was used to produce single-molecule maps and de novo assemble maps into a genome map with default parameters. Hybrid Scaffolds were assembled by hybrid Scaffold pipeline from Bionano Solve software package with default parameters.

**Hi–C**. We used Hi–C data to detect the chromosome contact information for assisting genome assembly. After crosslinking, the samples were used for quality control. Hi–C library preparation and sequencing using Illumina HiSeq platform with 2 × 150-bp reads at the Annoroad Gene Technology Co. Ltd. (Supplementary Table 6; Supplementary Fig. 15). Cleaned reads were first aligned to the reference genome using the bowtie2 end-to-end algorithm[57]. Unmapped reads are mainly composed of chimeric fragments spanning the ligation junction. According to the Hi–C protocol and the fill-in strategy, Hi–C-Pro (V2.7.8)[58] was used to detect the ligation site using an exact matching procedure and to align back on the genome the 5′ fraction of the read. The results of two mapping steps are then merged in a single alignment file. The assembly package, Lachesis, was applied to do clustering, ordering and orienting. We cut the chromosomes which predicted by Lachesis into bins with equal length such as 1 Mb or 500Kb and constructed heatmap based on the interaction signals that revealed by valid mapped read pairs between bins (Supplementary Fig. 15).

**Protein-coding gene annotation**. We used OMIGA[13] to annotate the codling moth genome by integrating evidence from homolog searching, transcriptome sequencing, and de novo predictions. Sequences of homologous proteins were downloaded from the NCBI invertebrate RefSeq. The transcriptome assembly was used to provide gene expression evidence which was assembled followed the protocol described by Trapnell[59]. Three ab initio gene prediction programs, including Augustus (version 3.1)[60], SNAP (version 2006-07-28)[61], and GeneMark-ET (Suite 4.21)[62] were used for de novo gene prediction. To obtain high accuracy, de novo gene prediction software must be retrained. We selected the transcripts with intact open reading frame (ORF) from the transcriptome to re-train Augustus and SNAP classifiers. To determine the transcripts with intact ORF, we used the BLAST search against the UniProtKB/Swiss-Prot proteins database (E-value = 1e−5) and Pfam to identify protein domains. After filtered by TransDecoder software, only the transcripts with a complete ORF were included. If genes had multiple transcripts, only the longest transcript was remained. Then, these gene transcripts were used to retrain the prediction software Augustus and SNAP. For GeneMark-ET, the whole assembly which more than 10 Mb were used to re-train the software. All gene evidence identified from above three approaches were combined by MAKER pipeline (version 2.31)[63] into a weighted and non-redundant consensus of gene structures. The default parameters were used for MAKER.

**Noncoding RNA gene annotation**. Three types of ncRNAs, transfer RNA (tRNA), rRNA, and small nuclear RNA, were annotated. To identify ncRNAs, the sequences of protein-coding genes, repetitive elements and other classes of noncoding RNAs were removed from the genome Scaffolds. tRNA genes were predicted by tRNAscan-SE[15] with eukaryote parameters. rRNA fragments were identified by aligning the rRNA template sequences from invertebrate animals to genomes using BLASTN with an E-value cutoff of 1E−5. Small nuclear RNA genes were inferred by the INFERNAL software against Rfam database of release 11.0[64]. The MapMi program (version 1.5.0)[65] was used to identify the miRNA homologs by mapping all miRNA matures in the miRBase[66] against the codling moth genome, and mirdeep2 software was used to identify novel miRNAs in the small RNA data. All algorithms were performed with default parameters.

**Detection of sex chromosomes**. Whole-genome alignments were generated using Satsuma with default values[67]. We compared sequencing coverage differences between male and female samples in order to detect sex-linked regions of the genome. Cytogenetic analysis reports substantial differentiation of the Z and W chromosome, thus we expect distinct patterns of Illumina sequencing coverage between sexes on the Z, W, and autosomes. Specifically, autosomes should have equal coverage while the Z should show an approximately two-fold greater coverage in males. The W should show a strongly female-biased coverage pattern, but the precise ratio is difficult to estimate because the W chromosome may contain regions of substantial sequence similarity to autosomes or the Z due either to shared repetitive sequences or homology to the neo-Z. The samples from the S population, providing three individuals of each sex, were aligned to the reference genome with bowtie. Read counts were tallied per scaffold, normalized by median sample coverage, and averaged by sex to give a single representative coverage value per scaffold for each sex. Additionally, scaffolds were similarly analyzed using nonoverlapping 500 bp windows in which to count and average reads and calculate male:female coverage.

**Receptor expression and voltage clamp recordings**. The receptor expression and two-electrode voltage clamp recordings were performed according to the previous works[68] with some modifications. The full-length coding sequences of *CpomOR3a*, *CpomOR3b*, and the co-receptor *CpomORco* (Genbank: JN836672.1) were amplified by PCR using the specific primers at both ends of ORFs, with carrying *Apa* I restriction site together with *Kozak* sequences in the forward primers and *Not* I restriction site in the reverse primers. The PCR products were digested with the both enzymes before ligation into PT7Ts vectors, which were previously linearized with the same enzymes. The cRNAs were synthesized from linearized vectors using mMESSAGE mMACHINE T7 Kit (Ambion, Austin, TX, USA). The cRNA mixture of 27.6 ng *CpomOrx* and 27.6 ng *CpomORco* was microinjected into the mature healthy oocytes (stage V–VII), which were previously treated with 2 mg/ml collagenase I in washing buffer (96 mM NaCl, 2 mM KCl, 5 mM MgCl$_2$, and 5 mM HEPES, pH 7.6) for 1–2 h at room temperature. After incubated for 4–7 days in incubation medium (1 x Ringer's buffer prepared with 0.8 mM CaCl$_2$ in washing buffer at pH 7.6, 5% dialyzed horse serum, 50 mg/ml tetracycline, 100 mg/ml streptomycin and 550 mg/ml sodium pyruvate) at 18 °C, the whole-cell currents against each chemical ($10^{-4}$ M in 1× Ringer's buffer) were recorded from the injected *X. oocytes* using a OC-725C two-electrode voltage clamp (Warner Instruments, Hamden, CT, USA) at a holding potential of -80 mV. The data were acquired and analyzed with Digidata 1440A and Pclamp10.0 software (Axon Instruments Inc., Union City, CA, USA). Column charts were generated using GraphPad Prism 5 (GraphPad software, San Diego, CA, USA). Statistics were carried out using IBM SPSS Statistics 22.0.0 (SPSS, Chicago, IL, USA).

**Genome resequencing**. To identify genetic changes conferring chemical insecticide resistance at genome level, two chemical insecticide resistant (Raz and Rv) and one chemical insecticide susceptible (S) strains provided by Dr. Pierre Franck and Dr. Myriam Siegwart of INRA (Avignon) were used in this study. Six third-instar larvae were randomly taken from each of the three strains, respectively. Total genomic DNA was isolated from the aforementioned 18 individuals, respectively. Genome of each individual was sequenced using the Illumina Hiseq 4000 platform at the Shenzhen Millennium Spirit Technology Co., Ltd.

**GWAS analysis**. To identify variants between chemical insecticide samples and the respective susceptible samples. Variants calling and association analysis for all resistant-susceptible samples comparison (RA–SV and RD–SV for insecticide resistance) were performed (Supplementary Fig. 16). The clean data of all samples were mapped to the genome assembly using BWA-mem[69] with default parameters. The overlapped reads in alignment were then removed by picard tools. Variants calling was performed between bam files of samples in each group by samtools[70] and bcftools[71]. Before the association analysis, variants stored in vcf files were filtered out by bcftools which removed variants with reads depth higher than 100 or quality less than 20% and by PLINK with the three thresholds: "--geno 0.05 --maf 0.01 --hwe 0.0001", which removed variants with missing genotype rates higher than 5%, minor allele frequency less than 1%, or Hardy–Weinberg equilibrium exact test p-value less than 0.001. Association analysis was performed between resistant strains and its corresponding susceptible strains by PLINK with the following parameters: --adjust --allow-extra-chr --allow-no-sex --assoc. Perl scripts were adopted to filter out the indel variants. To reduce the complexity of GWAS on identifying SNPs related to chemical insecticide resistance, we focused on the SNPs in 667 genes possibly involved chemical insecticide resistance from previous report[72]. Meanwhile, manhattan plot was drawn to visualize the SNPs located in cds regions in these 667 genes by qqman package of R[73].

**SNPs validation**. Ten individuals from each of the original three strains (S, Raz, and Rv,) reared in INRA were used for SNP validation. Insects from a laboratory strain rearing in the Institute of Plant Protection, Chinese Academy of Agricultural Sciences was used for RNA interference. The strain originated from a field codling moth population collected in 2013 in Gansu Province of China, and was reared on artificial diet in the laboratory at 24 ± 1 °C, 70% relative humidity and 16:8 h (L: D). Eleven SNPs which were significant different between the chemical insecticide resistant and susceptible sample were further confirmed in the individuals from the original strains by PCR. The PCR primers were designed according the sequences obtained. Ten individuals from S, Raz, and Rv were used to check each of the SNPs, respectively (Supplementary Table 23).

**RNA Interference**. RNAi was used to analyze the role of insecticide detoxifying of a P450 genes (ID: CPOM05212.t1, referred as CYP6B2) with the same significant SNPs between chemical insecticide resistance and susceptible strains, as well as to test the function of CpomOR3a/b. Sequence-specific primers target the CYP6B2 and CpomOR3a/b (Supplementary Table 21) were designed, and the siRNAs were chemically synthesized by Shanghai Gene Pharma (Shanghai, China) with 2′ Fluoro dU modification to increase the stability of the siRNAs. The siGFP was synthesized and used as a control. The siRNAs and siGFP were dissolved with nuclease-free water to the concentration of 2 μg/μl and stored at −80 °C until use. For CYP6B2 gene analyses, because all individuals of Raz and Rv strains were dead in 2018, we chose the Jiuquan strain which were used for de novo genome sequencing for function analysis. To knockdown CYP6B2, 0.5 μl siRNA was injected into the hemolymph of each forth-instar larva of Jiuquan strain using a microinjector (Femtojet Express, Eppendorf, Hamburg, Germany). The larvae injected with the same amount of siGFP and larvae had no injection were used as controls. Larvae were reared on artificial diet for 48 h post injection at 24 ± 1°C, 70% relative humidity and 16:8 h (L:D) until bioassay. For CpomOR3a/b gene functional test, 1 μl siRNA/siGFP was injected into the 9-day old pupae through the membrane. Moth will emerge from the survival pupae within 24 h post injection of CpomOR3a/b.

**Electroantennogram tests**. Electroantennogram tests were adopted from previous works[74]. Antennae were processed following standard procedures by cutting both extremes of flagella and immediately mounted with two glass capillary Ag/AgCl electrodes containing Ringer solution[75]. Pear ester solutions were loaded on a filter paper piece at the same dosages with y-tube tests. At least ten individuals were used as replicates for each chemical from each strain. Hexane was used as the carrier solvent and the blank control. Data were standardized following a standard protocol for EAG tests before compared between RNAi strains with siGFP strain by Student's t tests[76].

**Y-tube olfactometer assays**. Y-tube olfactometer indoor assays were adopted from our previous works on Lepidoptera adults[77]. The attractiveness of chemical volatiles was tested with 1-day-old adults. Pear ester was used at the dose of 1 mg. The choice made within 5 min was recorded and at least 30 moths were tested in each pair. All tests were conducted at room temperature, i.e., 25 ± 2 °C, with constant purified and moistened air flow at a rate of 0.5 l/min, and odorant compounds were switched between the two arms every fifth test. Chi-square tests were used to compare the differences of counts' distributions between siGFP strain and each other injected strain.

**Insecticide bioassay**. After 48 h post injection, thirty survival larvae from each treatment were randomly collected for each bioassay, and thirty fourth-instar native larvae without any injection were used as control. Three independent replicates were performed for each treatment and control. A droplet of 0.04 μl insecticide solution was applied topically on the middle-abdomen notum of the larvae with a hand microapplicator (Burkard Manufacturing Co. Ltd., Richmansworth England)[45]. A droplet of 0.10 μl of the LC50 solution of azinphos methyl (103.50 mg/L) and deltamethrin (3.55 mg/L) and imidacloprid (35.35 mg/L) in distilled water containing 0.01% (v/v) Triton and 0.01% acetone was applied topically on the middle-abdomen notum of the larvae with a hand microapplicator (Burkard Manufacturing, Richmansworth, England). Control larvae were treated with distilled water containing 0.01% (v/v) Triton X-100 and 0.01% acetone. Survival rate of the treated larvae were assessed in 48 h after exposure to the chemicals. Survival rate data (percentage) were transformed using arcsine square-root transformation, and then subjected to ANOVA. All ANOVA was analyzed by Tukey's Honest significant difference using GraphPad Prism 6.0 (GraphPad Prism Software Inc., San Diego, USA).

**Reporting summary**. Further information on research design is available in the Nature Research Reporting Summary linked to this article.

## Data availability
The sequence data from the Cydia genome project have been deposited in the GenBank under the accession number GCA_003425675.2. The BioProject of the Cydia genome project is PRJNA464426 and WGS project is QFTL02. The BioSample used for genome sequencing is SAMN09205828. The genome resequencing data of resistant strains have been deposited in the GenBank under SRR8479443-SRR8479460 and the transcriptome data have been deposited in SRA under SRR8479433-SRR8479442. The source data underlying Figs. 3c, 4a–d, 6d–f, and Supplementary Fig. 14 are provided as a Source Data file. All data mentioned in this paper can also be accessed at www.insect-genome.com/cydia/. All other relevant data is available upon request.

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

## Acknowledgements

We thank Professor Johannes A. Jehle of the Institute for Biological Control, Julius Kühn Institute, German for critical discussion, and B.F.A. Xiao-Qian Bao for technical support in producing the insect graphics. This work was supported by the National Key Research and Development Project of China (2016YFC1200602, 2017YFC1200602, and 2016YFC1201200), National Natural Science Foundation of China (31672033) and China's donation to the CABI Development Fund. The funders had no role in study design, data collection, and analysis, or in the decision to publish or in preparing the paper.

## Author contributions

F.L. conceived and designed the works of the whole paper. F.W. and F.L. coordinated the *Cydia* genome consortium. J.W. organized the researches on chromosomes. N.Y.

organized the researches on the expression and function analysis of the ORs and P450 genes. R. Tang did the research on the OR3 duplication. M.C. carried out the research on insecticide resistance. W.Q. and Y.X. prepared the samples for BioNano and Hi–C, did the sequencing and data analysis. W.Q. hosted the discussions in Shenzhen Genomics Institute of CAAS. J.S. and Jianyang G. reared the Jiuquan strain of codling moth. J.L. and C.Y. performed genome survey and K-mer analysis. K.H., C.Y., and X.Z. performed the flow cytometry analysis. C.Y. performed the genome annotation. C.Y. and X.Z. performed the comparative genomics analysis. L.X. did the chromosome location analysis. A.G., M.L., and J.W. performed the chromosome analysis. J.W., A.G., F.M., P.N., and M.D. did analysis on the Z and W chromosomes. C.H., R. Tang performed OR gene family analysis. G.W., E.J., R. Tang, W.L., C.M., and M.G. participated in OR analysis and discussion. C.H. and E.J. discovered the OR3 duplication. L.C., B.L., and W.F. did the GWAS analysis. Jinmeng G. did RACE to amplify the P450 genes. Jinmeng G., J.L., X.Z., R. Tian, performed the bioinformatics analysis of insecticide resistance associated genes. G.S., and Q.W. did the RNAi injections. C.J. did the RNA extraction and transcription. Y.X., G.S., and J.J. did qPCR analysis. L.L., W.K., and X.P. did the chemical insecticide bioassays. H.Z., C.J. and Q.W. did the Y-tube and EAG assays after the RNAi of OR3. R. Tang and Q.W. did statistical analysis of RNAi data. P.F., M.S., J.O., and S.M. provided samples of resistant trains and did the qPCR confirmation of CYP6B2 expression. C.Y., O.R., M.B., L.O., and G.A. did the comparative phylogenomic analysis. N.Y., F. Liu, S.W., X.X., G.Z., W.L., Y.X., Q.W., S.L., and M. Ye did the reference mining of the codling moth and made the figure of insect distribution. K.L, L.F., M.Y. and R.X., participated in resistance analysis. N.Y. drafted the manuscript. F.L., J.W., N.Y., M.C., R. Tang, E.J., and F.M. edited and improved the paper. F.L. and J.W. finalized the paper.

## Additional information

**Competing interests:** The authors declare no competing interests.

Fanghao Wan[1,2,23], Chuanlin Yin[3,23], Rui Tang[4,5,23], Maohua Chen[6,23], Qiang Wu[1,23], Cong Huang[1,7,23], Wanqiang Qian[2], Omar Rota-Stabelli[8], Nianwan Yang[1], Shuping Wang[9], Guirong Wang[1], Guifen Zhang[1], Jianyang Guo[1], Liuqi (Aloy) Gu[10], Longfei Chen[3], Longsheng Xing[2], Yu Xi[2], Feiling Liu[3], Kejian Lin[1], Mengbo Guo[1], Wei Liu[1], Kang He[3], Ruizheng Tian[6], Emmanuelle Jacquin-Joly[11], Pierre Franck[12], Myriam Siegwart[12], Lino Ometto[8,13], Gianfranco Anfora[8,14], Mark Blaxter[15], Camille Meslin[11], Petr Nguyen[16,17], Martina Dalíková[16,17], František Marec[16], Jérôme Olivares[12], Sandrine Maugin[12], Jianru Shen[1], Jinding Liu[18], Jinmeng Guo[18], Jiapeng Luo[3], Bo Liu[2], Wei Fan[2], Likai Feng[19], Xianxin Zhao[3], Xiong Peng[6], Kang Wang[6], Lang Liu[6], Haixia Zhan[4], Wanxue Liu[1], Guoliang Shi[1,20], Chunyan Jiang[1,20], Jisu Jin[1,7], Xiaoqing Xian[1], Sha Lu[1,20], Mingli Ye[21], Meizhen Li[3], Minglu Yang[22], Renci Xiong[22], James R. Walters[10] & Fei Li[3]

[1]State Key Laboratory for Biology of Plant Diseases and Insect Pests, Institute of Plant Protection, Chinese Academy of Agricultural Sciences, Beijing 100193, China. [2]Agricultural Genomics Institute at Shenzhen, Chinese Academy of Agricultural Sciences, Shenzhen 518120, China. [3]Ministry of Agriculture Key Lab of Molecular Biology of Crop Pathogens and Insect Pests, Institute of Insect Science, College of Agriculture and Biotechnology, Zhejiang University, Hangzhou 310058, China. [4]MARA-CABI Joint Laboratory for Bio-safety, Institute of Plant Protection, Chinese Academy of Agricultural Sciences, Beijing 100193, China. [5]State Key Laboratory of Integrated Management of Pest Insects and Rodents, Institute of Zoology, Chinese Academy of Sciences, Beijing 100101, China. [6]Northwest A&F University, State Key Laboratory of Crop Stress Biology for Arid Areas, Key Laboratory of Integrated Pest Management on Crops in Northwestern Loess Plateau of Ministry of Agriculture, Yangling 712100, China. [7]College of Plant Protection, Hunan Agricultural University, Changsha 410128, China. [8]Department of Sustainable Agro-ecosystems and Bioresources, IASMA Research and Innovation Centre, Fondazione Edmund Mach, Via Mach 1, 38010 San Michele all'Adige (TN), Italy. [9]Technical Centre for Animal Plant and Food Inspection and Quarantine, Shanghai Custom, Shanghai 200135, China. [10]Ecology and Evolutionary Biology, University of Kansas, Lawrence, KS 66046, USA. [11]INRA, Institute of Ecology and Environmental Sciences of Paris, 78000 Versailles, France. [12]INRA, Plantes et Systèmes de culture Horticole, 228 route de l'Aérodrome, 84914 Avignon Cedex 09, France. [13]Department of Biology and Biotechnology, University of Pavia, 27100 Pavia, Italy. [14]Centre Agriculture Food Environment (C3A), University of Trento, 38010 San Michele all'Adige (TN), Italy. [15]Edinburgh Genomics, and Institute of Evolutionary Biology, School of Biological Sciences, The King's Buildings, The University of Edinburgh, Edinburgh EH9 3JT, UK. [16]Biology Centre of the Czech Academy of Sciences, Institute of Entomology, Branišovská 31, 37005 České Budějovice, Czech Republic. [17]Faculty of Science, University of South Bohemia, Branišovská 1760, 37005 České Budějovice, Czech Republic. [18]College of Plant Protection, Nanjing Agricultural University, Nanjing 210095, China. [19]Institute of Plant Protection, Xinjiang Academy of Agricultural and Reclamation Sciences, Shihezi 832000, China. [20]College of Plant Health and Medicine, Qingdao Agricultural University, Qingdao 266109, China. [21]College of Biological and Environmental Engineering, Zhejiang Shuren University, Hangzhou 310015, China. [22]Xinjiang Production & Construction Corps Key Laboratory of Integrated Pest Management on Agriculture in South Xinjiang, Tarim University, Alar 843300, China. [23]These authors contributed equally: Fanghao Wan, Chuanlin Yin, Rui Tang, Maohua Chen, Qiang Wu, Cong Huang

