## [Peer Review File · Nature Communications]

Reviewers' comments:

Reviewer #1 (Remarks to the Author):

Review of the manuscript "The genetic basis of globalisation and destructiveness of codling moth, *Cydia pomonella*.

Wan et al describe a significant body of work on one of the most significant pests of horticulture worldwide. They have sequenced, assembled and annotated the genome of *Cydia pomonella* to produce a high quality chromosomal level assembly. They then go on to describe two pieces of functional genomics research, one on chemical ecology and a second on insecticide resistance. In both cases they use their genome assembly to discover new genes involved in these traits and provide evidence based on RNAi and other functional assays to demonstrate the role of a particular odorant receptor and p450 in chemical ecology and resistance to pesticides, respectively.

This work will provide a high quality genome assembly of an important group of moths that includes many pests (Tortricidae) and has significantly advanced the fields of both chemical ecology and insecticide resistance in insects.

I have only some minor suggestions and a number of suggested improvements to the wording of the manuscripts. As such I strongly recommend this manuscript for publication in Nature Communications.

Minor points:

Line 137: I presume the 42 females were all adults?

You should use the genera names in full the first time you use them in figure legends in case the figure is used in another publication.

Alternate title suggestion? I think this better reflects what you have actually achieved.

"A chromosome level assembly of the genome of the codling moth (*Cydia pomonella*) provides insights into its chemical ecology and insecticide resistance mechanisms."

When you talk about globalisation do you really mean invasiveness?

Line 312: Would be good to know the amino acid and nucleotide identity between the OR3 paralogues up front. If the identity is high how did you mitigate algorithms such as bowtie getting the mapping incorrect (check the methods). Might be good to reinforce that the RT-PCR is gene-specific by adding gene-specific before "reverse transcriptase-PCR". In your resequenced moths did you check to make sure the OR3 duplication is fixed and not segregating with some moths having one copy and some two?

Line 327: I'm not quite sure what you mean by divergent regulatory potentials. Do you simply mean that because they are separate genes they can evolve distinct expression patterns across tissues and development. If so you could simply refer to their potential to evolve new roles (neofunctionalisation) simply by changing expression. Although I note their ligand specificity is not identical so it looks like they are also undergoing neofunctionalisation by changing their ligand specificity presuming due to coding region changes.

Line 374: What is the evidence that OR3a and 3b are not expressed in sensilla tricoidea as well as sensilla auricillica. Is this a possibility? Or is OR1 the codlemone receptor in sensilla tricoidea?

I also think you can make more of the fact that the detection of mates and sites to oviposit seem to be linked within a single receptor able to detect compounds involved in both in the case of OR3. Certainly it looks as though pheromone receptors have evolved from receptors involved in detecting plant volatiles.

When you talk about 5 prime and 3 prime this should be annotated 5' and 3', not 5' and 3'.

Where did the 10⁻⁴ significance level come from in the GWAs screen. This seems quite low to me give the limited power you have in this analysis.

There is no validation of the genome assembly expect for comparing of two methods. Is it possible for you to provide any further confidence in the assembly? For example BAC sequencing or similar?

Minor wording suggestions:

Line 74: insert "have" after "We"

Line 79: move "without linkage mapping" to the end of the line

Line 83: kairomone and pheromone could be plural ... kairomones and pheormones

Line 83: delete "both"

Line 92: replace "capacities" with "capabilities"

Line 114: replace "about" with "regarding"

Line 116: delete "the"

Line 117: insert "have" before "mainly"

Line 117: move "unfortunately" to before "has"

Line 120: change to "and might have contributed to ..."

Line 129: insert "the" before "worldwide"

Line 130: change to "distribution of many insects"

Line 134: change to "long-read sequencing data, with scaffold assembly informed by both ..."

Line 137: change to "which was established from a collection from Jiuquan ..."

Line 146: insert "of" after "Gb"

Line 163: change to "two versions of the genome assembly, revealing that the two versions ..."

Line 191: change "lepidopterans" to "lepidopteran genes"

Line 194: change "ribosome" to "ribosomal"

Line 201: insert "a" after inferred

Line 205: replace "showing" with "estimating"

Line 208: New suggested first sentence for the figure legend for Figure 1. "Genomic characterisation and comparative genomics of the codling moth, *Cydia pomonella*."

Line 214: Should "ortholog" be "orthologous"? Same for Line 223?

Line 218: Insert "the" before "PhyloBayes"

Line 219: "node" should be "nodes"

Line 250: should "sequence" be plural? Same with line 261?

Line 268: "However, cytogenic data indicate that the W chromosome ..."

Line 274: "would" to "could"

Line 300: should you mention that the transcriptome is from antennae?

Line 329: should "fold" be "folded" in this Figure legend?

Line 343: rather than talking about "lines" perhaps "rows" is a better term.

Line 332: "indicate" to "indicates"

Line 333: "different" looks as though it has a space in it?

Line 339: the fact you have conducted some RNAseq seems rather buried in the manuscript. I think you been to be explicit about what new transcriptomes you have undertaken and where they have

been deposited.

Line 374: "raised" to "raise"

Line 375: "pheromone specific" to "pheromone-specific"

Line 383: "were" to "are"

Line 401: insert "levels of" before "resistance"

Line 411: Not sure what you mean by "increased enhancement of p450 enzyme activities"? Do you mean greater hydrolytic activity by p450s.

Line 413: Would be good to state the total number of p450 genes (146) in the text and not just in the figure legend.

Line 413: "showed" to "shows"

Line 414: change "and" to "with"

Line 415: change "had" to "having"

Line 419: insert "genes" before "was"

Line 426: insert "the" before "genome"

Line 436: "showed" should be "shows"

Line 453: insert "a" after "(mAChR)"

Figure 6d. You could consider using a log scale rather than an interrupted linear scale

Line 457: change "to" to "in"

Line 463: insert "the" before "susceptible"

Line 464: change "showed" to "given"

Line 466: "significant" to "significantly"

Line 467: delete "which"

Line 468: insert "higher" after "fold" in both instances on this line

Line 469: change "in" to "of"

Line 469: delete "showing that this"

Line 471: change "WT indicated the ..." to "WT indicates individuals ..."

Line 474: change "with" to "of"

Line 483: insert "of" after "69"

Line 492: suggest "To verify that CYP6B2 expression levels of indeed linked ..."

Line 504: delete "the"

Line 504: change "Taken" to "Taking"

Line 505: change "showed" to "demonstrates"

Line 509: change "yielding" to "to yield"

Line 513: insert "a" after "such"

Line 529: change "works" to "research has"

Line 530: change "was" to "has been"

Line 534: delete second "the"

Line 535: delete "thus"

Line 535: could use "spread" instead of "globalisation"

Line 542: insert "the" before "rapid"

Line 543: delete "the"

Line 560: Suggested last sentence ... " ... future genetics studies on the adaptation of codling moth populations to global agriculture changes and support the development of sustainable strategies of pest control."

Line 580: replace "in" with "of the"

Line 582: replace "on development of" with "in producing the"

Line 590: insert "the" before "whole"

Line 590: delete "works"

Line 591: replace "works" with "research"

Line 591: insert "the" before "OR3"

Line 592: replace "works" with "research" in the first instance and then "work" in the second.

Line 592: replace "function" with "functional"
Line 593: "of OR" to "of the ORs"
Line 594: "discussions" to "discussion"
Line 598: insert "the" before "comparative" and "chromosome"
Line 599: insert "the" before "chromosome"
Line 600: insert "the" before "Z"
Line 603: insert "the" before "GWAS" and "P450"
Line 603: change "amplifying" to "amplify"
Line 604: insert "the" before "bioinformatics"
Line 605: change "injection" to "injections"
Line 607: insert "the" after "did"
Line 607: change "bioassay" to "bioassays"
Line 607: insert "the" before "Y-tube"
Line 607: change "assay" to "assays"
Line 607: insert "the" before "RNAi"
Line 609: insert "the" before "qPCR"
Line 609: change "confirm" to "confirmation"
Line 610: insert "the" before "comparative"
Line 611: insert "the" before "reference"
Line 613: change "draft" to "drafted"
Line 614: do you mean "edited" rather than "wrote", given that the initial draft was written by N.Y.?

Richard Newcomb

Reviewer #2 (Remarks to the Author):

In MS# NCOMMS-19-04302-T ("The genetic basis of globalization and destructiveness of codling moth, *Cydia pomonella*") by Li et al, the authors conduct in-depth sequencing of a globally important pest, the codling moth, *Cydia pomonella*. The manuscript largely focuses on the sequencing and associated methodologies of this important pest, although the authors leverage functional testing of three identified genes.

Although the methods are robust, the manuscript largely focuses on the genomic sequencing methods to such a degree that the manuscript reads like a methods paper. In parallel, although the authors selectively test some of the identified genes, only three genes are tested and these results are distinct relative to the rest of the manuscript. In other words, the sub-headings are not well-linked, thus making the manuscript read like three or four different studies, rather than a single important manuscript. This significantly lessens my enthusiasm for the manuscript. Beyond this comment, there are a number of major and minor comments that are detailed below.

1. Related to the comments above, detailed methods take up a large part of the "Results" section, such that Lines 132-206 describe little of the results. It's unclear in this section what novel results are

presented. For instance, how do the results compare to other lepidopteran species in terms of chromosomal number or sex chromosomes? Although Table 1 shows these results, there is relatively little discussion about these differences. Similarly,

2. Sub-heading connection. The sequencing and olfactory receptor sub-headings could be better linked. The OR section is largely based on previous studies in this domain, and it's unclear what new information the new genomic data provide. What other ORs show strong selection? What others show strong expression levels? What other ORs could candidates for control interventions, and how does the new genomic data provide this insight? These questions are lacking in this section, making it distinct compared to the previous sections.

3. Genetic basis of insecticide resistance. Again, as this section is written, it appears that the authors largely leverage previously published results (51-53) in this section, and find similar results as previous studies(53,54). It's unclear what new functional insights are provided by the genomic studies.

4. There are a number of grammatical and formatting errors in the manuscript. These are detailed below:

- a. Line 72: Re-write for clarity/grammar.
- b. Lines 80-84: Re-write for clarity/grammar.
- c. Line 101: Replace "uncontrolled" for clarity.
- d. Lines 111-113: Change for grammar / clarity.
- e. Lines 147-168 should be shortened.
- f. Sub-heading "Synteny, karyotype and sex chromosomes": This is an interesting section, but given the functional analysis in the other major subsections, it raises the question about the functional aspects or testing of these results.
- g. Lines 226-229: Place into first subheading.
- h. Line 297: Change word "exploits" to "detects" or another descriptor.
- i. Lines 300-303: The previous studies really raises the question what new information this subsection brings. Moreover, it is poorly linked with the previous sections.
- j. Figure 3: Show data for all ORs, rather than Or3a and b.
- k. Lines 529-531: Rewrite for grammar and clarity.
- l. References: Please check references for proper formatting.

Reviewer#1's comments (Richard Newcomb)

Wan et al describe a significant body of work on one of the most significant pests of horticulture worldwide. They have sequenced, assembled and annotated the genome of *Cydia pomonella* to produce a high quality chromosomal level assembly. They then go onto to describe two pieces of functional genomics research, one on chemical ecology and a second on insecticide resistance. In both cases they use their genome assembly to discover new genes involved in these traits and provide evidence based on RNAi and other functional assays to demonstrate the role of a particular odorant receptor and p450 in chemical ecology and resistance to pesticides, respectively.

This work will provide a high quality genome assembly of an important group of moths that includes many pests (Tortricidae) and has significantly advanced the fields of both chemical ecology and insecticide resistance in insects.

I have only some minor suggestions and a number of suggested improvements to the wording of the manuscripts. As such I strongly recommend this manuscript for publication in Nature Communications.

Response: We greatly appreciate your time and expertise in reviewing our manuscript. We have carefully revised the manuscript according to your constructive comments, which significantly improve our manuscript. Thanks again for your positive enthusiasm for our work.

Minor points:

Line 137: I presume the 42 females were all adults?

Response: Yes, they were adults. We now indicate this in the revision.

You should use the genera names in full the first time you use them in figure legends in case the figure is used in another publication.

Response: corrected.

Alternate title suggestion? I think this better reflects what you have actually achieved. "A chromosome level assembly of the genome of the codling moth (*Cydia pomonella*) provides insights into its chemical ecology and insecticide resistance mechanisms."

Response: It is a great suggestion, and we change the title to "A chromosome-level genome assembly of the codling moth (*Cydia pomonella*) provides insights into its chemical ecology and insecticide resistance".

When you talk about globalization do you really mean invasiveness?

Response: Yes, we mean invasiveness. This has been corrected.

Line 312: Would be good to know the amino acid and nucleotide identity between the OR3 paralogues up front. If the identity is high how did you mitigate algorithms such as bowtie getting the mapping incorrect (check the methods). Might be good to reinforce that the RT-

PCR is gene-specific by adding gene-specific before “reverse transcriptase-PCR”. In your re-sequenced moths did you check to make sure the OR3 duplication is fixed and not segregating with some moths having one copy and some two?

Response: The OR3a and OR3b share 94% identity in nucleotide and 89% identity in amino acids. Thus, it is feasible to distinguish these two paralogues when using bowtie to map the reads to scaffolds. We revise the methods to reinforce that RT-PCR is gene-specific. We confirmed that in all re-sequenced moths, the OR3 duplication is fixed.

Line 327: I’m not quite sure what you mean by divergent regulatory potentials. Do you simply mean that because they are separate genes they can evolve distinct expression patterns across tissues and development. If so you could simply refer to their potential to evolve new roles (neofunctionalisation) simply by changing expression. Although I note their ligand specificity is not identical so it looks like they are also undergoing neofunctionalisation by changing their ligand specificity presuming due to coding region changes.

Response: Changed as suggested.

Line 374: What is the evidence that OR3a and 3b are not expressed in sensilla tricoidea as well as sensilla auricillica. Is this a possibility? Or is OR1 the codlemone receptor in sensilla tricoidea?

Response: Our FISH experiments showed that OR3a and 3b are expressed in sensilla auricillica. In our experiments, we did not see any labelling in sensilla trichodea, although we cannot completely exclude this. This topic is interesting and deserves further studies, for instance co-labelling of OR1 with sensilla trichodea markers. Nevertheless, we have revised this section to better clarify the current evidence pertaining to sensilla-specific expression and implications thereof.

I also think you can make more of the fact that the detection of mates and sites to oviposit seem to be linked within a single receptor able to detect compounds involved in both in the case of OR3. Certainly, it looks as though pheromone receptors have evolved from receptors involved in detecting plant volatiles.

Response: We have updated this section of the manuscript to reflect this suggestion.

When you talk about 5 prime and 3 prime this should be annotated 5' and 3', not 5’ and 3’.

Response: Corrected, thank you.

Where did the 10⁻⁴ significance level come from in the GWAs screen. This seems quite low to me given the limited power you have in this analysis.

Response: We did a reference mining for GWAS cutoff and found that the cutoff 10⁻⁴ is commonly used. Here is the table for GWAS *P*-value. So, we keep the cutoff as 10⁻⁴ in this work.

Table 1 The commonly used cutoff for GWAS analysis

Organism	P value	Reference
Human	1×10^{-4}	Aragam, et al. Journal of Molecular Neuroscience , 2013, 50(2): 250-256.
maize	1×10^{-4}	Samayoa, et al. BMC plant biology , 2015, 15(1): 35.
Arabidopsis thaliana	5×10^{-8}	Genomes Consortium. Electronic address, et al. Cell , 2016, 166(2): 481-491.
Arabidopsis thaliana	1×10^{-4}	Kloth, et al. Journal of experimental botany , 2016, 67(11): 3383-3396.
Arabidopsis	1×10^{-4}	Proietti, et al. Plant, cell & environment , 2018, 41(10): 2342-2356.
Rice	1×10^{-4}	Dingkuhn, et al. Journal of experimental botany , 2017, 68(15): 4369-4388.
Rice	1×10^{-4}	Kadam, et al. Journal of experimental botany , 2018, 69(16): 4017-4032.
Drosophila melanogaster	1×10^{-5}	Duneau, et al. G3: Genes, Genomes, Genetics , 2018, 8(11): 3469-3480.
Fagus grandifolia Ehrh	1.585×10^{-5}	Calic, et al. BMC genomics , 2017, 18(1): 547.
Human	1×10^{-6}	Liu, et al. Cell , 2018, 175(2): 347-359. e14.
Friesian horse	1.63×10^{-7}	Schurink, et al. BMC genetics , 2018, 19(1): 49.
Human	1×10^{-8}	Yengo, et al. Journal of Molecular Neuroscience , 2013, 50(2): 250-256.

There is no validation of the genome assembly expect for comparing of two methods. Is it possible for you to provide any further confidence in the assembly? For example, BAC sequencing or similar?

Response: It is difficult to find BAC sequencing service in China now, so we have used additional data from Nanopore and PacBio sequencing platforms to validate the genome assembly quality (please see responses to Editor comments for further details). We added the following data and analyses:

1) Sequencing the genomic DNA with Oxford Nanopore platform.

In total, we obtained ~71 Gb Nanopore long reads with mean length greater than 23 Kb; the longest read is 223 Kb (Table 1). Aligning the reads to the reference genomes showed that 99.96% reads can be mapped to the genome assembly. There are 6,070 reads whose length is larger than or equal to 100 Kb, and these ultra-long reads can be uniquely mapped to the genome scaffolds with high consistency, suggesting the high reliability of the genome assembly.

Table 2 The sequencing data statistic of Nanopore sequencing

Feature	Number
of_total_Reads_Bases (bps)	83,473,696,842
of_total_Reads_Number	4,161,465
Pass_Reads_Bases (bps)	71,105,727,881
Pass_Reads_Number	3,068,220
Pass_Reads_Mean_Length (bps)	23174.91
Pass_Reads_N50_Length (bps)	32,637
Pass_Reads_Medium_Length (bps)	18,322
Pass_Reads_Max_Length (bps)	223,241

2) Sequencing full-length transcripts with PacBio platform.

For the full-length transcript sequencing, totally 37.57 Gb with 704,348 polymerase reads were obtained, yielding 500,583 full-length non-chimeric (FLNC) circular consensus sequence (CCS) subreads with mean length 2343,32 bps. We finally got 25,940 high quality consensus isoform transcripts and 15,260 protein coding transcripts with complete open read frame (ORF), with the mean lengths of 2,571.98 and 1,239.45 bp, respectively (Table 2). More than 93% full-length transcripts can be exclusively mapped to the reference genome.

Table 3 The statistics of the full-length transcripts mapped to reference genome

Feature	Number
Total Reads	25,940
Mapped Reads	24,326
Mapping Rate	0.937779
UnMapped Reads	1,614
MultiMap Reads	782
MultiMap Rate	0.030146
Reads Mapping Forward	11,822
Reads Mapping Reverse	11,722

We greatly appreciate the careful reading and supportive editing of our manuscript that is reflected in this thorough list of corrections.

Minor wording suggestions:

Line 74: insert “have” after “We”

Response: Corrected

Line 79: move “without linkage mapping” to the end of the line

Response: Corrected

Line 83: kairomone and pheromone could be plural ... kairomones and pheormones

Response: Corrected

Line 83: delete “both”

Response: Corrected

Line 92: replace “capacities” with “capabilities”

Response: Corrected

Line 114: replace “about” with “regarding”

Response: Replaced with “concerning”.

Line 116: delete “the”

Response: Corrected

Line 117: insert “have” before “mainly”

Response: Corrected

Line 117: move “unfortunately” to before “has”

Response: Corrected

Line 120: change to “and might have contributed to ...”

Response: Corrected

Line 129: insert “the” before “worldwide”

Response: Corrected

Line 130: change to “distribution of many insects”

Response: Corrected

Line 134: change to “long-read sequencing data, with scaffold assembly informed by both ...”

Response: Corrected

Line 137: change to “which was established from a collection from Jiuquan ...”

Response: Corrected the whole sentence.

Line 146: insert “of” after “Gb”

Response: Corrected

Line 163: change to “two versions of the genome assembly, revealing that the two versions ...”

Response: Corrected

Line 191: change “lepidopterans” to “lepidopteran genes”

Response: Corrected

Line 194: change “ribosome” to “ribosomal”

Response: Corrected

Line 201: insert “a” after inferred

Response: Corrected

Line 205: replace “showing” with “estimating”

Response: Corrected

Line 208: New suggested first sentence for the figure legend for Figure 1. “Genomic characterisation and comparative genomics of the codling moth, *Cydia pomonella*.”

Response: Corrected

Line 214: Should “ortholog” be “orthologous”? Same for Line 223?

Response: Corrected

Line 218: Insert “the” before “PhyloBayes”

Response: Corrected

Line 219: “node” should be “nodes”

Response: Corrected

Line 250: should “sequence” be plural? Same with line 261?

Response: Corrected

Line 268: “However, cytogenic data indicate that the W chromosome ...”

Response: Corrected

Line 274: “would” to “could”

Response: Corrected

Line 300: should you mention that the transcriptome is from antennae?

Response: Corrected

Line 329: should “fold” be “folded” in this Figure legend?

Response: Change the “fold” to “angled”.

Line 343: rather than talking about “lines” perhaps “rows” is a better term.

Response: Corrected

Line 332: “indicate” to “indicates”

Response: Corrected

Line 333: “different” looks as though it has a space in it?

Response: Corrected

Line 339: the fact you have conducted some RNAseq seems rather buried in the manuscript. I think you been to be explicit about what new transcriptomes you have undertaken and where they have been deposited.

Response: The transcriptome data have been deposited in SRA under SRR8479433-SRR8479442.

Line 374: “raised” to “raise”

Response: Corrected

Line 375: “pheromone specific” to “pheromone-specific”

Response: Corrected

Line 383: “were” to “are”

Response: Corrected

Line 401: insert “levels of” before “resistance”

Response: Corrected

Line 411: Not sure what you mean by “increased enhancement of p450 enzyme activities”? Do you mean greater hydrolytic activity by p450s.

Response: Corrected

Line 413: Would be good to state the total number of p450 genes (146) in the text and not just in the figure legend.

Response: Corrected

Line 413: “showed” to “shows”

Response: Corrected

Line 414: change “and” to “with”

Response: Changed to two sentences.

Line 415: change “had” to “having”

Response: Corrected

Line 419: insert “genes” before “was”

Response: Deleted the sentence.

Line 426: insert “the” before “genome”

Response: Corrected

Line 436: “showed” should be “shows”

Response: Corrected

Line 453: insert “a” after “(mAChR)”

Response: Corrected

Figure 6d. You could consider using a log scale rather than an interrupted linear scale

Response: figure 6d updated

Line 457: change “to” to “in”

Response: Corrected

Line 463: insert “the” before “susceptible”

Response: Corrected

Line 464: change “showed” to “given”

Response: Corrected

Line 466: “significant” to “significantly”

Response: Re-wrote the sentence.

Line 467: delete “which”

Response: Re-wrote the sentence.

Line 468: insert “higher” after “fold” in both instances on this line

Response: Corrected

Line 469: change “in” to “of”

Response: Corrected

Line 469: delete “showing that this ...”

Response: Corrected

Line 471: change “WT indicated the ...” to “WT indicates individuals ...”

Response: Corrected

Line 474: change “with” to “of”

Response: Corrected

Line 483: insert “of” after “69”

Response: Corrected

Line 492: suggest “To verify that CYP6B2 expression levels of indeed linked ...”

Response: Corrected

Line 504: delete “the”

Response: Corrected

Line 504: change “Taken” to “Taking”

Response: Corrected
Line 505: change “showed” to “demonstrates”
Response: Corrected
Line 509: change “yielding” to “to yield”
Response: Corrected
Line 513: insert “a” after “such”
Response: Corrected
Line 529: change “works” to “research has”
Response: Corrected
Line 530: change “was” to “has been”
Response: Corrected
Line 534: delete second “the”
Response: Corrected
Line 535: delete “thus”
Response: Corrected
Line 535: could use “spread” instead of “globalisation”
Response: Corrected
Line 542: insert “the” before “rapid”
Response: Corrected
Line 543: delete “the”
Response: Corrected
Line 560: Suggested last sentence ... “ ... future genetics studies on the adaptation of codling moth populations to global agriculture changes and support the development of sustainable strategies of pest control.”
Response: Corrected
Line 580: replace “in” with “of the”
Response: Corrected
Line 582: replace “on development of” with “in producing the”
Response: Corrected
Line 590: insert “the” before “whole”
Response: Corrected
Line 590: delete “works”
Response: Corrected
Line 591: replace “works” with “research”
Response: Corrected
Line 591: insert “the” before “OR3”
Response: Corrected
Line 592: replace “works” with “research” in the first instance and then “work” in the second.
Response: Corrected
Line 592: replace “function” with “functional”
Response: Corrected
Line 593: “of OR” to “of the ORs”
Response: Corrected
Line 594: “discussions” to “discussion”
Response: Corrected
Line 598: insert “the” before “comparative” and “chromosome”
Response: Corrected
Line 599: insert “the” before “chromosome”

Response: Corrected

Line 600: insert “the” before “Z”

Response: Corrected

Line 603: insert “the” before “GWAS” and “P450”

Response: Corrected

Line 603: change “amplifying” to “amplify”

Response: Corrected

Line 604: insert “the” before “bioinformatics”

Response: Corrected

Line 605: change “injection” to “injections”

Response: Corrected

Line 607: insert “the” after “did”

Response: Corrected

Line 607: change “bioassay” to “bioassays”

Response: Corrected

Line 607: insert “the” before “Y-tube”

Response: Corrected

Line 607: change “assay” to “assays”

Response: Corrected

Line 607: insert “the” before “RNAi”

Response: Corrected

Line 609: insert “the” before “qPCR”

Response: Corrected

Line 609: change “confirm” to “confirmation”

Response: Corrected

Line 610: insert “the” before “comparative”

Response: Corrected

Line 611: insert “the” before “reference”

Response: Corrected

Line 613: change “draft” to “drafted”

Response: Corrected

Line 614: do you mean “edited” rather than “wrote”, given that the initial draft was written by N.Y.?

Response: Corrected

3. Reviewer #2’s comments (Remarks to the Author):

In MS# NCOMMS-19-04302-T ("The genetic basis of globalization and destructiveness of codling moth, *Cydia pomonella*") by Li et al, the authors conduct in-depth sequencing of a globally important pest, the codling moth, *Cydia pomonella*. The manuscript largely focuses on the sequencing and associated methodologies of this important pest, although the authors leverage functional testing of three identified genes.

Although the methods are robust, the manuscript largely focuses on the genomic sequencing methods to such a degree that the manuscript reads like a methods paper.

Response: We greatly appreciate your time and expertise in reviewing our manuscript. Thanks for your keen eyes and constructive criticism. We concur that the original version of our manuscript contained an excess of methodological detail, and we have worked to substantially streamline this aspect of the manuscript, moving many such details to the

methods or supplemental sections, and retaining only the most significant methodological details.

In parallel, although the authors selectively test some of the identified genes, only three genes are tested, and these results are distinct relative to the rest of the manuscript. In other words, the sub-headings are not well-linked, thus making the manuscript read like three or four different studies, rather than a single important manuscript. This significantly lessens my enthusiasm for the manuscript. Beyond this comment, there are a number of major and minor comments that are detailed below.

Response: Thank you for pointing out that we had not fully succeeded in generating a manuscript that sufficiently integrates the various research efforts presented here into a coherent research narrative. This sentiment resonates with the editor's suggestion to elaborate on the links between the genomic and functional analysis. With the aim to improve this aspect of the manuscript, we have reworked the text to better integrate the different parts of this manuscript to make a more coherent and consistent final product. We decided that sub-heading titles we used were too broad and thus changed the sub-heading titles in revision. We have chosen to focus on three genes of particular interest to highlight significant characteristics of the codling moth evolutionary ecology.

In this work, we present a chromosome-level genome assembly for the codling moth, which served as an essential foundation for discovering novel genetic aspects of noteworthy eco-physiological features of codling moth. As we detail in our responses to the Editor's comments (see above), the gene duplication of OR3, which contributes to enhanced abilities for locating food and mates, could not be confidently identified without a high quality assembly. Additionally, through the GWAS analysis, we identified three SNPs in the promoter of CYP6B2 and further confirmed their role in conferring insecticide resistance. Our perspective is that the genome assembly was an essential component of these functional investigations concerning the OR3 duplication and insecticide resistance, and we hope our revisions to the manuscript better capture this integrative nature of our research efforts.

1. Related to the comments above, detailed methods take up a large part of the "Results" section, such that Lines 132-206 describe little of the results. It's unclear in this section what novel results are presented. For instance, how do the results compare to other lepidopteran species in terms of chromosomal number or sex chromosomes? Although Table 1 shows these results, there is relatively little discussion about these differences.

Response: As mentioned above, we removed large parts of methods information in results. We also now discuss chromosome number difference and sex chromosomes between lepidopteran insects. We developed the discussion about the comparison of *Cydia pomonella* genome with those of other Lepidoptera species (Line 157-161, Line 500-504).

2. Similarly, Sub-heading connection. The sequencing and olfactory receptor sub-headings could be better linked. The OR section is largely based on previous studies in this domain, and it's unclear what new information the new genomic data provide. What other ORs show strong selection? What others show strong expression levels? What other ORs could candidates for control interventions, and how does the new genomic data provide this insight? These questions are lacking in this section, making it distinct compared to the previous sections.

Response: We provide additional explanations and discussions about their connections with genome data.

3. Genetic basis of insecticide resistance. Again, as this section is written, it appears that the authors largely leverage previously published results (51-53) in this section, and find similar results as previous studies (53,54). It's unclear what new functional insights are provided by the genomic studies.

Response: we are sorry we did not introduce our results clearly. We cited reference 51-53 to show that the cluster of P450 genes has been widely observed in lepidopteran. References 53 and 54 are the papers introducing the resistant strains we used in this work. The novel insights we present are the detection of hundreds of SNPs significantly differentiating susceptible and resistant strains which might confer insecticide resistance in codling moth populations. Except for two previously known mutations involved in pyrethroid and organophosphorus resistances in the sodium channel (reference 50) and in acetylcholinesterase (reference 55) genes, respectively, none of these SNPs have been reported before. Moreover, we validated the roles of three SNPs in the promoter of CYP6B2 conferring insecticide resistance to several chemical insecticides. These three SNPs and the functional role of CYP6B2 in insecticide resistance were not reported before. It is a very important discovery for our understanding of insecticide resistance in codling moth populations worldwide, at least in Europe and China.

It is true that it is well known that CYPs are involved in pesticides detoxification in many insects, including the codling moth. Here, we identified a gene among the huge number of CYPs that is functionally involved in the resistance to chemical insecticides in the codling moth, which opens up possibilities to easily monitor worldwide insecticide resistance in codling moth populations using simple genetic markers. We have made additions to the text to highlight this point (Line 439-443).

4. There are a number of grammatical and formatting errors in the manuscript. These are detailed below:

Response: Thanks for the close eyes on our manuscript. We have an international team and some authors are native speakers in English, and these authors have now carefully polished the English writing in the manuscript. Additionally, reviewer 1 also generously provided extensive suggestions and corrections to improve clarity and precision in writing. We are grateful for all of this input to improve the quality of our manuscript.

a. Line 72: Re-write for clarity/grammar.

Response: Corrected

b. Lines 80-84: Re-write for clarity/grammar.

Response: Corrected

c. Line 101: Replace "uncontrolled" for clarity.

Response: Corrected

d. Lines 111-113: Change for grammar / clarity.

Response: Corrected

e. Lines 147-168 should be shortened.

Response: We remove those sentences describing methods.

f. Sub-heading "Synteny, karyotype and sex chromosomes": This is an interesting section, but given the functional analysis in the other major subsections, it raises the question about the functional aspects or testing of these results.

Response: We appreciate this section is of notable interest. Our motivation in these analyses stems primarily from questions of evolutionary and comparative genomics, and our primary aim was to characterize the pattern/history of biodiversity in codling moth, both relative to

other Lepidoptera as well as between chromosomes in the genome. We see this as an important (and interesting) contribution towards fully contextualizing the data provided in the genome assembly. As such, we view these results as an end unto themselves, and not as a means towards subsequent functional analysis. Accordingly, we do not clearly comprehend the nature of “the question” concerning function or testing to which you allude, but do not explicitly articulate. We apologize we cannot better intuit what you are asking for here, and we hope that, in light of clarifying our motivation, this element of the manuscript can satisfactorily stand on its own.

g. Lines 226-229: Place into first subheading.

Response: Corrected

h. Line 297: Change word "exploits" to "detects" or another descriptor.

Response: Re-wrote these sentences.

i. Lines 300-303: The previous studies really raises the question what new information this subsection brings. Moreover, it is poorly linked with the previous sections.

Response: Only one copy of *OR3* has been previously found in an antennae transcriptome. The new information this subsection brings is: with the assistance of a high-quality genome assembly, we found two copies of this gene. The duplication of this gene forms a tandem repeat on chromosome 17. We further confirmed that this duplication significantly improves the ability of codling moth to detect pear ester. We rewrite this section to clarify this.

j. Figure 3: Show data for all ORs, rather than Or3a and b.

Response: Provided and updated

k. Lines 529-531: Rewrite for grammar and clarity.

Response: Changed

l. References: Please check references for proper formatting.

Response: Corrected

REVIEWERS' COMMENTS:

Reviewer #1 (Remarks to the Author):

The revised manuscript addresses all the review suggestions for improvement.

Reviewer #2 (Remarks to the Author):

The authors have done an excellent job of addressing my earlier comments. The manuscript reads well, and the sections are well-integrated. Moreover, the importance of the study is now readily apparent and in my opinion, it should be well-cited and an important study.

REVIEWERS' COMMENTS:

Reviewer #1 (Remarks to the Author):

The revised manuscript addresses all the review suggestions for improvement.

Reviewer #2 (Remarks to the Author):

The authors have done an excellent job of addressing my earlier comments. The manuscript reads well, and the sections are well-integrated. Moreover, the importance of the study is now readily apparent and in my opinion, it should be well-cited and an important study.

Response: we greatly appreciate two reviewers' time and expertise.